# Kolmogorov-Arnold Hierarchical Implicit Neural Representation Model for Physical Field Reconstruction

## Abstract

Reconstructing continuous fields from sparse observations poses one of the most persistent challenges in scientific machine learning, with critical implications for understanding geophysical phenomena from limited sensor networks. Although implicit neural representations (INRs) have recently emerged as promising solutions, capturing fine-scale structures in complex domains such as atmospheric and oceanic systems remains elusive. We introduce KHINR (Kolmogorov-Arnold Hierarchical Implicit Neural Representation) that achieves state-of-the-art spatial field reconstruction through a fusion of learnable Gabor filters and Kolmogorov-Arnold Network (KAN) blocks in a hierarchical structure. The sparse spatial data points are first encoded using learnable Gabor filters to extract localized, frequency-aware spatial features that are further processed in the latent space via a hierarchical structure with KAN blocks. For reconstruction, the Gabor-encoded unknown spatial points are passed through a gating mechanism on the latent representation learned by the hierarchical KAN blocks. During training only the sparse data is used, without the full field data. Rigorous evaluation across four distinct physical fields from meteorological and ocean datasets reveals KHINR's superior performance compared to other leading models on multiple reconstruction tasks under varying sparsity conditions. Comprehensive ablation studies validate the critical contribution of each architectural component, establishing KHINR as a new standard for sparse-to-continuous field reconstruction in scientific applications.

## 1 Introduction

The reconstruction of continuous physical fields such as temperature, sea level, biological tracers from sparse data is a fundamentally under-determined problem, presenting challenges in various scientific and engineering contexts (Fukami et al., 2021; Shen et al., 2015; Zhang et al., 2018). Sparse sensors are often the most practical and efficient means of data collection, especially in disciplines such as geophysics (Reichstein et al., 2019), astronomy (Gabbard et al., 2022), biochemistry (Zhong et al., 2021), and fluid mechanics (Deng et al., 2023).

A variety of traditional and machine learning methods are used to reconstruct continuous dense fields from sparse observations. Traditional approaches to reconstructing dense fields are based on physics-based models grounded in partial differential equations (PDEs) derived from fundamental conservation laws and physical principles (Hughes, 2003). Although these methods offer interpretability and a strong theoretical foundation, they are computationally demanding and require extensive modeling based on first principles when dealing with complex high-dimensional systems such as weather and epidemiology (Brunton et al., 2016; Massucci et al., 2016). Moreover, integrating real-world observational data into these PDE-based frameworks for calibration and validation presents significant challenges (Raissi et al., 2019). To address the limitations of traditional methods, machine learning methods such as Voronoi-based convolutional neural networks (Fukami et al., 2021), ensemble self-training (Zhang et al., 2024), neural radiance fields (Mildenhall et al., 2021), occupancy networks (Mildenhall et al., 2021), and variants of physics-informed neural networks (Zhang et al., 2024; Smith et al., 2022; Santos et al., 2022) have been proposed. Appendix B has more information on related work. However, compared to conventional image and video data, sci-

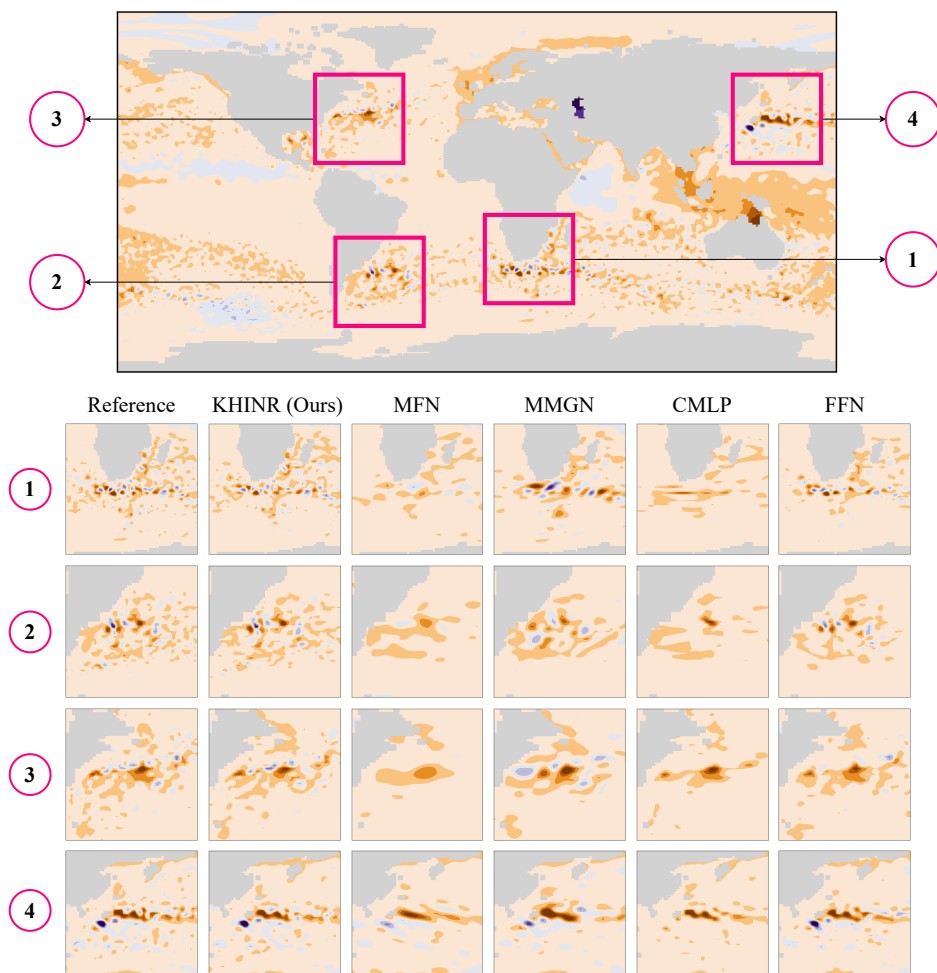

Figure 1: Reconstruction of sea surface height (SSH) from sparse observations using different implicit neural representations. The top panel shows the global reference Sea Surface Height (SSH) with four highlighted regions (1-4) shown below in a detailed comparison of reconstruction across different methods A single INR is trained for the global data in all cases. Our KHINR provides the best reconstruction with complex details in all regions that other leading models fail to capture.

entific data that characterize complex physical systems present distinctive challenges for machine learning. Specific challenges include learning multiscale dynamics (Stachenfeld et al., 2021) from sparse and irregular observations from mobile sensors (Rodrigues et al., 2021), often have limited spatial coverage (Myers & Schultz, 2000), and may go offline randomly due to operational difficulties (Paletta et al., 2022). Furthermore, achieving scalability and generalization to new data remains difficult.

These constraints underscore the need for adaptable and data-driven methods capable of learning continuous functions from limited datasets without over dependence on explicit physical models. Implicit neural representations (INRs) have become a potent framework for learning continuous spatial signals directly from spatio-temporal coordinates as input. Unlike grid/mesh-based methods, INRs naturally accommodate sparse and irregular sensor data by learning basis functions that represent spatial and temporal changes (Luo et al., 2024). Crucially, INRs require only discrete observations for training, negating the need for continuous data streams.

Recent advances in INRs can be categorized based on improvements in four key areas: activation functions, positional encoding, combined techniques, and network structure enhancement (Essakine

et al., 2024). In the activation function segment, recent advancements enhance the expressiveness and versatility of INRs by optimizing activation functions, resulting in improved representations. For example, SIREN (Sitzmann et al., 2020a), WIRE (Saragadam et al., 2023), GAUSS (Ramasinghe & Lucey, 2022b), HOSC (Serrano et al., 2024), SINC Saratchandran et al. (2024), and FINER (Liu et al., 2024a) demonstrate unique benefits in signal modeling through specialized activations. Within the category of advances in positional encoding, techniques such as Fourier features (Mescheder et al., 2019) focus on refining positional data encoding to better capture complex signals. Among approaches that integrate activation functions and positional encoding, Trident (Shen et al., 2023) shines by tackling both components at once, providing a more robust and adaptable strategy for representation learning. Lastly, within the network structure category, methods such as Incode (Kazerouni et al., 2024), MFN (Fathony et al., 2020), and FR (Shi et al., 2024), target the optimization of the network architecture as a whole to improve learning and generalization.

Existing INR methodologies, while promising, encounter significant challenges in reconstructing physical fields: *(i)* traditional MLPs lack the structural biases needed to effectively model the multiscale and anisotropic properties of scientific phenomena, leading to inefficient sampling and poor scalability at high resolutions, *(ii)* current positional encodings, such as Fourier features (Tancik et al., 2020b) and sinusoidal activations (Sitzmann et al., 2020b), depend on global basis functions that fail to accurately represent spatially varying frequencies and orientation-dependent patterns in physical fields, thus becoming vulnerable to noise and ill-suited to adapt to local signal features, and *(iii)* missing mechanisms that leverage global structural dependencies while retaining computational efficiency hinder coherent reconstruction of physical phenomena across various spatial scales.

We propose a new hierarchical implicit neural representation that effectively addresses the above limitations through two main innovations:

1. We integrate Hierarchical Kolmogorov-Arnold Networks (KANs) within INRs, which utilize learnable univariate functions. This endows our model with multiscale representational abilities and improved inductive biases for modeling physical fields.

2. A latent cross-attention mechanism is introduced to compress global structural dependencies into a lower-dimensional embedding space, allowing scalable reconstruction and maintaining crucial physical relationships across scales.

In addition, we replace global positional encodings with shift-invariant Gabor filters to provide localized adaptive frequency representations similar to WIRE and MFN (Saragadam et al., 2023; Fathony et al., 2020). Figure 1 illustrates how this combination enables our model to accurately capture complex details of the physical field, outperforming existing INR techniques. In what follows, we introduce our novel architecture and present various experiments and ablation studies that highlight the importance of each architectural component in enhancing reconstruction performance.

## 2 METHODOLOGY

### 2.1 PROBLEM STATEMENT

Let $\mathbf{v}(\mathbf{k}, t)$ represent the sparse sensor measurements of the physical fields collected at discrete spatial locations $\mathbf{k} \in \mathbb{R}^{d_x}$ and time points $t$. The purpose of field reconstruction is to learn a continuous function $u(\mathbf{x}, t)$ that accurately represents the physical field at any spatial coordinate $\mathbf{x}$ and time $t$. Reconstructing $u(\mathbf{x}, t)$ from sparse observations is challenging due to high dimensionality and inherent nonlinear, multi-scale dynamics of physical processes.
Our objective is to develop an INR architecture $\hat{u}(\mathbf{x}, t)$ that is trained using only sparse observations $\mathbf{v}(\mathbf{k}, t)$ to accurately reconstruct complex physical fields. In contrast to physics-informed neural networks and traditional data assimilation, which require physics priors from first principles or explicit PDE constraints, our approach learns directly from the sparse observational data alone.

### 2.2 PROPOSED FRAMEWORK

As discussed in Section 1, coordinate-based INRs for approximating continuous physical fields suffer from spectral bias, reliance on global basis functions, and insufficient inductive bias. We address these through three key innovations: (1) **Learnable Gabor filters** (Section 2.2.2) provide local-

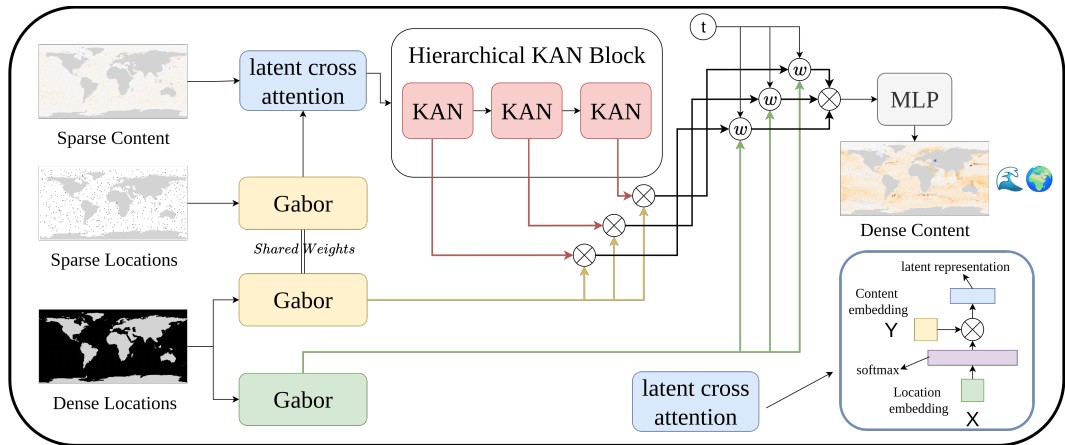

Figure 2: **KHINR inference architecture.** Sparse observation locations ($\mathbf{k}$) and content ($\mathbf{v}(\mathbf{k})$) are encoded through Gabor filters and latent cross-attention. Dense query locations ($\mathbf{x}$) are similarly Gabor-encoded. Hierarchical KAN blocks (red) extract multi-scale features, which are fused with spatial and temporal information through multiplicative gating and an output MLP to produce reconstructed values $\hat{u}(\mathbf{x})$. During training dense locations and dense content are replaced with sparse locations and sparse content (See Figure 5).

ized, adaptive frequency representations; (2) **Hierarchical Kolmogorov-Arnold Networks** (Sections 2.2.1, 2.3) use learnable univariate functions that naturally capture multi-scale dependencies; (3) **Gabor-enhanced latent cross-attention** (Section 2.3.1) compresses global dependencies into lower-dimensional space for computational efficiency.

**Architecture overview (Figure 2):** The model processes sparse observation locations $\mathbf{k}$, sparse content $\mathbf{v}(\mathbf{k})$, and dense query locations $\mathbf{x}$. Coordinates are encoded using Gabor filters, sparse content undergoes latent cross-attention, hierarchical KAN blocks extract multi-scale features, and a multiplicative gating mechanism fuses information streams to reconstruct values at query locations. Training uses only sparse locations and content instead of dense locations and dense content.

### 2.2.1 KOLMOGOROV-ARNOLD NETWORKS

The Kolmogorov-Arnold representation theorem (Hecht-Nielsen, 1987) states that any multivariate continuous function $f : [0, 1]^n \to \mathbb{R}$ can be expressed as:

$$f(x_1, \ldots, x_n) = \sum_{q=1}^{2n+1} \Phi_q \left( \sum_{p=1}^{n} \phi_{q,p}(x_p) \right),$$

where $\phi_{q,p} : [0, 1] \to \mathbb{R}$ and $\Phi_q : \mathbb{R} \to \mathbb{R}$ are continuous univariate functions. KANs generalize this by substituting each weight parameter with a learnable univariate function (Liu et al., 2024b). For a deep KAN: $\text{KAN}(\mathbf{x}) = (\Phi_{L-1} \circ \Phi_{L-2} \circ \cdots \circ \Phi_0)(\mathbf{x})$, where the $j$-th feature in layer $l$ transforms as:

$$x_{l,j} = \sum_{i=1}^{n_{l-1}} \Phi_{l-1,j,i}(x_{l-1,i}), \quad \Phi(x) = w_a \cdot \text{SiLU}(x) + w_b \cdot \text{Spline}(x),$$

where $\text{Spline}(x) = \sum_i c_i B_i(x)$ is a linear combination of B-spline functions with learnable coefficients. Unlike MLPs with fixed activations, KANs provide adaptive nonlinearities at each connection, improving sample efficiency and multi-scale representation critical for sparse observations of physical fields.

### 2.2.2 GABOR-BASED COORDINATE ENCODING

Traditional positional encodings, such as Fourier features, employ global basis functions that struggle to capture spatially varying frequency content. We leverage localized Gabor filters combining

Gaussian spatial localization with sinusoidal frequency selectivity:

$$g(\mathbf{x}) = \exp\left(-\frac{\gamma}{2}\|\mathbf{x} - \boldsymbol{\mu}\|_2^2\right) \times \sin(\mathbf{W}_g\mathbf{x} + \mathbf{b}_g),$$

where $\boldsymbol{\mu} \in \mathbb{R}^{d_h}$ (filter center), $\gamma \in \mathbb{R}^{d_h \times d_x}$ (spatial extent), $\mathbf{W}_g \in \mathbb{R}^{d_x \times d_x}$ (frequency matrix) and $\mathbf{b}_g \in \mathbb{R}^{d_x}$ (phase bias) are learnable. During training, $\boldsymbol{\mu}$ clusters around regions with high variability; larger $\gamma$ values create localized filters for fine details, while smaller values capture global patterns; $\mathbf{W}_g$ adapts to dominant frequencies; $\mathbf{b}_g$ aligns the oscillations with the signal characteristics. We initialize $\boldsymbol{\mu} \sim \mathcal{N}(2, 1)$ in the spatial domain, $\gamma \sim Gamma(\alpha, \beta)$, $\mathbf{W}_g \sim \mathcal{N}(0, \beta^2)$, and $\mathbf{b}_g \sim U[-\pi, \pi]$.

### 2.3 ARCHITECTURE

#### 2.3.1 GABOR-ENHANCED LATENT CROSS-ATTENTION

To address quadratic scaling of attention with spatial resolution, we compress sparse observations into fixed-size latent space. Let $\mathbf{k}_s \in \mathbb{R}^{N_s \times d_x}$ denote sparse locations and $\mathbf{v}_s \in \mathbb{R}^{N_s \times d_c}$ content values. The next steps are as follows.

1. Project content: $\mathbf{V} = \mathbf{v}_s\mathbf{W}_c$, where $\mathbf{W}_c \in \mathbb{R}^{d_c \times d_h}$
2. Encode locations: $\mathbf{Z}_{\text{loc}} = \text{Gabor}(\mathbf{k}_s) \in \mathbb{R}^{N_s \times d_h}$
3. Use learnable queries: $\mathbf{Q} = \mathbf{Z}_{\text{loc}}\mathbf{W}_q$, where $\mathbf{Z}_{\text{loc}} \in \mathbb{R}^{L \times d_h}$ (typically $L = 256\text{-}512$)
4. $Q^{'} = \textbf{Softmax}(Q)$ and $V^{'} = W_v.\text{LayerNorm}(V)$
5. Output: $\mathbf{L} = Q^{'} \times V^{'} \in \mathbb{R}^{L \times d_h}$

This reduces complexity from $O(N_s^2)$ to $O(L^2)$ for the subsequent learning stages, where $L \ll N_s$, enabling efficient processing. As shown in Figure 3, this reduces training time by 40–60% while maintaining accuracy. Figure 4 illustrates the architecture overview of Gabor-Enhance Latent Cross-Attention.

#### 2.3.2 HIERARCHICAL KAN BLOCKS

The latent representation $\mathbf{L}$ feeds through $K$ hierarchical KAN blocks, producing outputs $\{\mathbf{z}_s^{(k)} \in \mathbb{R}^{B \times L \times d_h}\}$. This hierarchical arrangement enables multi-scale feature learning, where each KAN layer processes information at different levels of abstraction: lower layers capture fine-grained spatial patterns while higher layers extract increasingly complex semantic relationships.

The hierarchical KAN framework is critical for three reasons. First, it serves as a learnable substitute for conventional multi-layer perceptrons, providing enhanced expressiveness with adaptable basis functions at each layer rather than fixed activations. Second, this sequential arrangement enables the network to incrementally refine its representations, effectively capturing both localized spatial patterns from Gabor-enhanced encodings and broader semantic contexts from the latent content representation. Third, this architecture adeptly captures intricate non-linear interactions essential for converting sparse geographic knowledge into dense content features, which is crucial for precise field reconstruction.

Our hierarchical structure allows independent multi-scale feature extraction. Different blocks specialize in different frequency ranges with multiplicative gating (Section 2.3.3) dynamically weights their contributions based on query location. Appendix F.4.3 shows this hierarchical configuration achieves 12–36% lower error than single-block or sequential architectures.

#### 2.3.3 GATING MECHANISM

The gating mechanism synthesizes coordinate-based spatial features, hierarchical semantic representations, and temporal context to produce reconstructions. For query location $\mathbf{x}$:

**Stage 1 – Latent Construction:** For each KAN block $k$, we compute channel-wise mean $\bar{\mathbf{z}}^{(k)} = \text{mean}_c(\mathbf{z}_s^{(k)}) \in \mathbb{R}^{B \times L \times 1}$ and modulate by shared Gabor-encoded query(yellow in Figure 2): $\mathbf{Z}^{(k)} = \bar{\mathbf{z}}^{(k)} \odot \mathbf{q}$, where $\mathbf{q} = \text{Gabor}(\mathbf{x})$.

**Stage 2 – Block-wise Fusion:** Each block $i$ combines three complementary information streams. The **coordinate term** $\text{Coord}_i = \text{Gabor}(\mathbf{x}).\mathbf{W}_g^{(i)}$ (green in Figure 2) encodes pure spatial patterns through learned coordinate filters, capturing geometric structure independent of content. The **latent term** $\text{Latent}_i = \mathbf{Z}^{(i)}\mathbf{W}_z^{(i)}$ incorporates semantic content from the hierarchical KAN representations, conditioning spatial reasoning on observed field values. The **temporal term** $\text{LatentTime}_i = \mathbf{L}_{\text{time}}\mathbf{W}_{\text{lat}}^{(i)}$ integrates global temporal context, enabling the model to adapt reconstruction based on time-dependent patterns. These three terms are combined additively: $\text{Out}_i = \text{Coord}_i + \text{Latent}_i + \text{LatentTime}_i + \mathbf{b}^{(i)}$, where $\mathbf{b}^{(i)}$ is a learned bias.

**Stage 3 – Multiplicative Fusion:** Unlike additive fusion, we use multiplicative integration $\text{Out}_{\text{final}} = \prod_{i=1}^{K} \text{Out}_i$. This enables complex non-linear interactions where each block can modulate contributions from others, implementing soft gating for scale-dependent reconstruction. Multiplicative fusion naturally captures higher-order interactions between scales, allowing conditional activation based on context.

**Stage 4 – Final Mapping:** $\hat{u}(\mathbf{x}) = \text{MLP}_{\text{out}}(\text{Out}_{\text{final}})$, where $\text{MLP}_{\text{out}}$ is a 2–3 layer network with SiLU activations.

### 2.3.4 Temporal Handling

For datasets with temporal context, time $t$ is encoded using learnable embeddings $\mathbf{t}_{enc}$, then projected to $\mathbf{L}_{\text{time}} = \mathbf{t}_{\text{enc}}\mathbf{W}_t$ for use in gating. This captures periodic temporal patterns (seasonal cycles) as a global context.

### 2.3.5 Training and Inference

**Training:** During training, the model learns to fit the sparse observations $\{(\mathbf{k}_i, v_i)\}_{i=1}^{N_s}$. We encode sparse locations via Gabor filters, process sparse content through latent cross-attention and hierarchical KAN blocks, then reconstruct values at the same sparse locations using the gating mechanism. We minimize L2 loss: $\mathcal{L} = \frac{1}{N_s}\sum_i \|\hat{u}(\mathbf{k}_i) - v_i\|_2^2$. See Figure 5 for architecture details during training.

**Inference (sparse-to-dense, Figure 2):** At inference, the trained model performs continuous field reconstruction. Given sparse observations $\{(\mathbf{k}_i, v_i)\}$ and dense query grid $\mathbf{X}$. The computational cost scales linearly with $N_{\text{dense}}$ rather than quadratically, allowing real-time high-resolution reconstruction (Figure 7).

## 3 Experiments

### 3.1 Datasets

To evaluate the performance of the model, we considered four challenging meteorological and ocean datasets derived from simulation and remote sensing.

**Remotely sensed altimeter derived sea surface height data (SSH):** We use Sea Level Anomalies (SLA) gridded data product, derived from multiple altimeter missions. The anomaly is calculated in relation to a 1993-2012 mean. These SLAs are determined using optimal interpolation, which combines the L3 along-track measurements (Copernicus Marine Service, 2024b). We use data for the year 2023, at a horizontal resolution of 1 degree. This data set includes 365 time steps, 180 latitudinal points and 360 longitudinal points.

**Sea surface chlorophyll analysis and forecast data (CHL):** We utilized the global chlorophyll mass concentration analysis and forecast product from the NEMO model that is assimilated with satellite chlorophyll (Copernicus Marine Service, 2024a). We use daily data from January 15, 2024, to August 15, 2025, interpolated at 1 degree spatial resolution. This dataset has 579 time steps, 170 latitude points (-80 to 90), and 360 longitude points.

**Climate simulation based global surface temperature data (GST):** We use the Community Earth System Model version 2 (CESM2) (Danabasoglu et al., 2020), a fully coupled global climate model.

Specifically, we used monthly averaged global surface temperature fields, representing an atmospheric variable, for model evaluation. To increase the complexity of the task, the seasonal cycle has been removed from the data. The resulting data set has dimensions of 1024 time steps, 192 latitudinal points, and 288 longitudinal points.

**Remotely sensed sea surface temperature data (SST):** The sea surface temperature data are sourced from a retrospective dataset with a four-day latency and a near-real-time dataset with a one-day latency (Martin et al., 2012). Wavelets are used as basis functions for optimal interpolation on a global grid with a resolution of 0.01 degrees. Our analysis focuses on the Gulf Stream region using one year of daily data, from August 20, 2022, to August 20, 2023, at the specified spatial resolution of 1/100th of a degree. The data set spans 360 time steps, 901 latitude points, and 1001 longitude points. Appendix D has further details of all datasets.

## 3.2 Implementation Details

We benchmarked our model against seven leading implicit neural networks: SIREN (Sitzmann et al., 2020a), FFN-G and FFN-P (Tancik et al., 2020a), MMGN (Luo et al., 2024), CMLP (Ramasinghe & Lucey, 2022a), MFN (Fathony et al., 2020), WIRE (Saragadam et al., 2023). Further details of the baseline models are in Appendix E.2.

We assessed the performance of our KHINR in various reconstruction tasks designed to test its robustness under different levels of sensor randomness and data sparsity. Four increasingly complex tasks were selected to evaluate how well the KHINR can manage randomness–**Task 1:** a fixed number of sensors at set locations; **Task 2:** a variable number of sensors at set locations; **Task 3:** a fixed number of sensors at random locations; and **Task 4:** a variable number of sensors at random locations. An illustrative figure explaining all four tasks can be found in Appendix E.4. Each task was further examined at three different sparsity levels to simulate partial observations: 5, 25, and 50% for GST; 0.1, 0.3, and 0.5% for SST; and 10, 25, and 50% for SSH and CHL. The evaluation involved comparing against the full ground truth (100% state) to gauge reconstruction accuracy.

The models were trained to minimize the L2-loss using 200 epochs with the AdamW optimizer (learning rate of 0.001 and weight decay of 1e-5). Batch sizes were set at 16, 2, 1, and 1 for GST, SST, SSH, and CHL, respectively. All experiments were executed using PyTorch Lightning on a single NVIDIA A6000 GPU (48GB of vRAM). Details of all hyperparameters are in Appendix E.

## 3.3 Results

**Quantitative Results.** Tasks 1 to 4 at three different ratios for GST, SST, SSH and CHL datasets are used to comprehensively evaluate KHINR vis-a-vis other leading methods. Tables 1 and 2 present quantitative evaluation results in terms of MSE for SSH, CHL, GST, and SST, confirming the efficacy of KHINR. Our model consistently outperforms baselines in all four datasets and various task setups. Its performance remains stable, regardless of dataset attributes or evaluation ratios, underscoring KHINR's robustness and adaptability. PSNR and SSIM metrics are provided in the Appendix G.1 to further strengthen the analysis.

**Qualitative Results.** Figure 1 showed the superior reconstruction fidelity of KHINR for the SSH dataset in Task 4 with a 25% sampling ratio. Other qualitative reconstruction plots for various sampling ratios and tasks are presented in the Appendix G.2.

## 4 Model Analysis

### 4.1 Hierarchical KAN Effect

Table 3 compares the effectiveness of Single Block, Sequential, and Hierarchical KAN architectures across multiple datasets (GST, SST, SSH) and tasks. The Hierarchical (2 blocks) setup consistently achieves the lowest error rates, demonstrating clear improvements over both the Single Block and Sequential (2 blocks) configurations. For example, in the SSH dataset (Task 4), Hierarchical KAN outperformed Sequential by yielding a relative promotion of over 36%. Similar trends are observed across GST and SST datsets, where Hierarchical KAN achieves up to 20% performance gains. These results highlight that structuring KANs hierarchically, rather than sequentially or as a

Table 1: Comparison of KHINR with INR models on SSH and CHL datasets at different sampling ratios for different tasks using MSE($10^{-3}$). Best results are in **bold**, second best are underlined.

| Model | SSH | | | | CHL | | | |
|---|---|---|---|---|---|---|---|---|
| | Task 1 | Task 2 | Task 3 | Task 4 | Task 1 | Task 2 | Task 3 | Task 4 |
| *Sampling ratio = 10%* | | | | | | | | |
| SIREN | 15.55 | 15.55 | 15.55 | 15.55 | 21.13 | 21.55 | 8.90 | 8.95 |
| FFN-P | 5.49 | 5.44 | 1.15 | **1.16** | 37.94 | 37.53 | 7.80 | **7.87** |
| FFN-G | 6.31 | 6.14 | 1.93 | 2.06 | 24.32 | 26.41 | 9.38 | 9.36 |
| MMGN | 17.36 | 13.90 | 3.04 | 3.88 | 29.88 | 27.66 | 9.65 | 10.27 |
| CMLP | 5.56 | 5.69 | 3.09 | 3.14 | 17.36 | 18.12 | 14.96 | 13.20 |
| MFN | 7.13 | 6.98 | 3.08 | 3.07 | 18.40 | 19.33 | 11.47 | 11.48 |
| WIRE | 8.09 | 8.09 | 8.09 | 8.09 | 57.17 | 57.36 | 52.26 | 52.99 |
| KHINR (ours) | **4.24** | **4.29** | **1.03** | 1.46 | **14.64** | **14.87** | **7.40** | 9.00 |
| *Sampling ratio = 25%* | | | | | | | | |
| SIREN | 15.55 | 15.55 | 15.55 | 15.55 | 10.71 | 12.18 | 7.89 | 8.33 |
| FFN-P | 3.96 | 3.94 | 0.79 | 0.82 | 17.81 | 17.93 | 7.12 | 7.27 |
| FFN-G | 3.52 | 3.73 | 1.26 | 1.37 | 11.76 | 13.11 | 8.57 | 8.62 |
| MMGN | 6.32 | 6.66 | 1.93 | 2.51 | 12.77 | 13.58 | 8.79 | 9.42 |
| CMLP | 4.01 | 4.27 | 4.52 | 4.68 | 13.24 | 14.21 | 13.93 | 11.86 |
| MFN | 3.76 | 4.15 | 2.87 | 2.83 | 12.81 | 13.75 | 10.89 | 11.04 |
| WIRE | 8.09 | 7.99 | 8.08 | 8.14 | 52.26 | 56.24 | 43.62 | 52.89 |
| KHINR (ours) | **2.89** | **2.89** | **0.46** | **0.72** | **9.51** | **10.40** | **4.23** | **6.03** |
| *Sampling ratio = 50%* | | | | | | | | |
| SIREN | 15.55 | 15.52 | 15.55 | 15.17 | 8.18 | 9.16 | 7.41 | 7.63 |
| FFN-P | 2.09 | 2.31 | 0.69 | 0.72 | 9.41 | 10.35 | 6.79 | 6.85 |
| FFN-G | 1.84 | 2.09 | 1.10 | 1.03 | 8.99 | 9.84 | 8.09 | 8.17 |
| MMGN | 2.17 | 2.97 | 1.64 | 1.97 | 9.05 | 10.23 | 8.46 | 9.06 |
| CMLP | 2.97 | 3.37 | 2.55 | 2.86 | 12.52 | 12.66 | 11.70 | 12.14 |
| MFN | 2.98 | 3.27 | 2.71 | 2.79 | 11.12 | 11.73 | 10.69 | 10.64 |
| WIRE | 8.09 | 7.83 | 8.09 | 7.90 | 41.17 | 53.68 | 46.36 | 52.39 |
| KHINR (ours) | **1.39** | **1.57** | **0.35** | **0.47** | **5.27** | **6.04** | **3.38** | **4.40** |

single block, significantly enhances their representational efficiency and robustness across diverse tasks. Ablation study showing the effectiveness of hierarchical compared with only the last layer representation across the GST, SST, and SSH datasets with sampling ratios of 5%, 0.3%, and 10%, respectively, for Task 1 and Task 4 with 2 layers. MSE is reported.

Table 4 compares standard MLP against hierarchical KAN in our KHINR model across GST, SST and SSH datasets for Tasks 1 and 4. Hierarchical KAN outperforms MLP in all scenarios. For the SST dataset with 0.3% sampling, hierarchical KAN reduces error by over 65% in Task 1 and 47% in Task 4 compared to MLP. Notable improvements occur in GST and SSH datasets, with gains ranging from 12% to 22%. These results demonstrate Hierarchical KAN's superior performance over standard MLPs, especially with sparse data. Additional ablations are in Appendix F.4.

## 4.2 GABOR EFFECT

We conducted an ablation study to demonstrate the behavior of different positional encodings: MLP, Fourier, and Gabor. We used GST sampled at 5% and SST sampled at 0.3% to study the effect. Table 5 shows that Gabor encoding results in the lowest MSE compared to MLP and Fourier-based encoding. The superior performance shows that the localized encoding by Gabor filters addresses spectral bias in standard MLPs by providing spatially-adaptive basis functions, enabling efficient learning of complex physical field patterns while maintaining robustness to noise and irregular sampling.

## 4.3 GABOR ENHANCED LATENT ATTENTION

Figure 3 illustrates the reduction in training time per epoch by employing Gabor-enhanced latent attention in various tasks and sampling ratios for SST, GST, SSH, and CHL. The inclusion of Gabor-enhanced latent attention (shown by orange bars) consistently decreases the training time compared to the baseline model without latent attention (shown by blue bars), without reducing model perfor-

Table 2: Comparison of KHINR with other baselines on GST and SST datasets at different sampling ratios for different tasks using MSE($10^{-3}$) metrics. Best results are in **bold**, second best are underlined.

| Model | GST | | | | SST | | | |
|---|---|---|---|---|---|---|---|---|
| | Task 1 | Task 2 | Task 3 | Task 4 | Task 1 | Task 2 | Task 3 | Task 4 |
| *Sampling ratio = 5% (GST), 0.1% (SST)* | | | | | | | | |
| SIREN | 15.36 | 15.84 | 13.64 | 13.63 | 1.24 | 1.53 | 0.72 | 0.83 |
| FFN-P | 9.37 | 10.15 | 7.67 | 7.60 | 2.51 | 2.91 | 0.70 | 0.83 |
| FFN-G | 18.03 | 19.65 | 14.11 | 14.12 | 8.33 | 9.04 | 1.09 | 1.58 |
| MMGN | 4.54 | 5.00 | 3.06 | 3.79 | 0.98 | 1.25 | **0.47** | 1.02 |
| CMLP | 20.15 | 20.76 | 19.59 | 18.98 | 2.03 | 2.19 | 1.69 | 1.56 |
| MFN | 19.45 | 20.78 | 15.82 | 15.85 | 10.51 | 9.98 | 1.53 | 1.69 |
| WIRE | 24.40 | 24.40 | 24.40 | 24.40 | 83.17 | 82.31 | 83.16 | 83.06 |
| KHINR (ours) | **3.69** | **4.44** | **2.56** | **3.50** | **0.87** | **1.12** | 0.49 | **0.80** |
| *Sampling ratio = 25% (GST), 0.3% (SST)* | | | | | | | | |
| SIREN | 12.27 | 12.26 | 12.39 | 12.44 | 0.54 | 0.74 | 0.35 | 0.40 |
| FFN-P | 6.83 | 7.15 | 6.61 | 6.51 | 0.73 | 0.96 | 0.33 | **0.37** |
| FFN-G | 13.59 | 13.46 | 12.96 | 13.26 | 1.09 | 2.32 | 0.49 | 0.89 |
| MMGN | 13.47 | 3.19 | 2.74 | 2.91 | 4.00 | **0.54** | 0.26 | 0.43 |
| CMLP | 18.98 | 18.54 | 18.46 | 18.57 | 1.53 | 1.62 | 1.49 | 1.55 |
| MFN | 15.46 | 15.82 | 15.07 | 15.18 | 2.49 | 3.45 | 1.03 | 1.20 |
| WIRE | 24.40 | 24.41 | 24.40 | 24.51 | 83.07 | 70.58 | 83.15 | 76.16 |
| KHINR (ours) | **2.01** | **2.41** | **1.92** | **2.27** | **0.34** | **0.54** | **0.21** | **0.37** |
| *Sampling ratio = 50% (GST), 0.5% (SST)* | | | | | | | | |
| SIREN | 12.28 | 12.32 | 12.04 | 12.21 | 0.35 | 0.45 | 0.26 | 0.32 |
| FFN-P | 6.71 | 6.78 | 6.50 | 6.56 | 0.46 | 0.58 | 0.29 | **0.31** |
| FFN-G | 13.05 | 13.00 | 12.71 | 13.02 | 0.68 | 0.88 | 0.55 | 0.58 |
| MMGN | 2.77 | 2.92 | 2.71 | 2.82 | 0.30 | 0.40 | 0.25 | 0.35 |
| CMLP | 18.54 | 18.37 | 17.97 | 17.65 | 1.26 | 1.43 | 1.42 | 1.40 |
| MFN | 15.16 | 15.23 | 15.12 | 15.11 | 1.47 | 1.81 | 1.08 | 1.05 |
| WIRE | 24.40 | 24.42 | 24.40 | 24.35 | 83.12 | 36.55 | 79.23 | 63.39 |
| KHINR (ours) | **1.91** | **2.17** | **1.87** | **2.12** | **0.23** | **0.38** | **0.19** | **0.31** |

Table 3: Ablation study showing the effectiveness of Hierarchical compared with only last layer representation across the GST, SST, and SSH datasets with sampling ratios of 5%, 0.3%, and 10%, respectively, for Task 1 and Task 4 with 2 layers. MSE($10^{-3}$) is reported.

| Dataset | GST | GST | SST | SST | SSH | SSH |
|---|---|---|---|---|---|---|
| | Task1 | Task4 | Task1 | Task4 | Task1 | Task4 |
| Sequential | 4.44 | 4.33 | 0.99 | 1.00 | 3.10 | 1.13 |
| Hierarchical | 3.69 | 3.50 | 0.87 | 0.80 | 2.89 | 0.72 |
| Promotion | 16.89% | 19.17% | 12.12% | 20.00% | 6.77% | 36.28% |

mance (Table 11 in Appendix F.3). In particular, in SST with sampling ratios of 0.3% and 0.5%, the baseline model takes twice the time compared to our model. GST exhibits similar trends, with the training time without Gabor-enhanced latent attention reaching 170 seconds at 50% ratio, while our model with Gabor-enhanced latent attention only requires around 33 seconds. For SSH and CHL tasks, the training time is reduced by almost half or more. These results highlight the efficiency gains achieved by integrating Gabor-enhanced latent attention, enabling faster training.

Table 4: Ablation study demonstrating the superiority of Hierarchical KAN over MLP across the GST, SST, and SSH datasets with sampling ratios of 5%, 0.3%, and 10%, respectively, for Task 1 and Task 4, evaluated using the MSE($10^{-3}$) metric.

| Dataset | GST | GST | SST | SST | SSH | SSH |
|---|---|---|---|---|---|---|
| | Task1 | Task4 | Task1 | Task4 | Task1 | Task4 |
| MLP | 4.40 | 4.28 | 0.98 | 0.70 | 5.42 | 1.66 |
| KAN | 3.69 | 3.50 | 0.34 | 0.37 | 4.24 | 1.46 |
| Promotion | 16.14% | 18.22% | 65.30% | 47.14% | 21.77% | 12.05% |

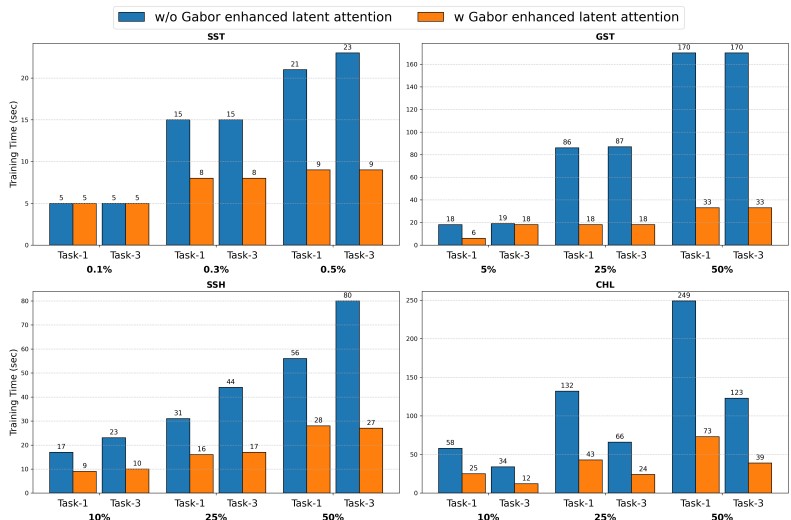

Figure 3: Comparison of training times for our model with and without Gabor-enhanced latent attention across various datasets in Task 1 and Task 3 using different sampling ratios

Table 5: Ablation study highlighting the effectiveness of Gabor filters over Fourier and MLP layers in KHINR on GST and SST datasets, with sampling ratios of $5\%$ and $0.3\%$, for Task 1 and Task 4, evaluated using the $\text{MSE}(10^{-3})$ metric.

| Positional Encoding | GST (Task 1) | GST (Task 4) | SST (Task 1) | SST (Task 4) |
|---|---|---|---|---|
| MLP | 4.57 | 5.11 | 0.55 | 0.85 |
| Fourier | 10.07 | 4.77 | 3.40 | 0.55 |
| Gabor | **3.69** | **3.50** | **0.34** | **0.37** |

In addition to these ablation studies, we also evaluated the efficiency and performance of the model through computational (App F.1) and statistical (App F.2) analyses. In particular, KHINR has the lowest inference time and the highest reconstruction rank (Figure 7). Extensive sensitivity studies have been performed by varying the latent dimension (App F.4.1), hidden dimension (App F.4.2), and the hierarchical blocks (App F.4.3) have been performed to demonstrate that the chosen architecture consistently yields the best trade-off between accuracy and computational cost.

## 5 CONCLUSION

KHINR introduces a new architecture for reconstructing physical fields from sparse data by combining hierarchical Kolmogorov-Arnold Networks with learnable Gabor filters and latent cross-attention. This architecture solves key problems that have limited existing implicit neural representations in scientific applications. The model consistently outperforms all baselines in four different datasets, showing strong performance even with extremely limited data. This capability directly addresses real-world challenges where sensors are expensive or difficult to deploy widely. The ability of KHINR to reconstruct fine-scale structures while maintaining computational efficiency opens unprecedented opportunities for real-time environmental monitoring, climate modeling, and sensor network optimization. KHINR's design shows the power of combining deep learning with first-principles understanding of the physical world, contributing to the next generation of physics-aware neural architectures.

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

## A    LLM USAGE

LLM based grammar check tool has been used to polish language after it was written by the authors.

## B    RELATED WORK

Continuous field reconstruction methods can be broadly categorized into two approaches: *(i)* classical and *(ii)* deep learning methods.

One of the first and most widely used classical methods is the linear model. To address non-linearity, spatial interpolation is performed using inverse distance weighting with a power parameter (Shepard, 1968). Later, alternative approaches are based on the Gaussian process and its variants (Rasmussen, 2003), such as the ensemble Kalman filter (Evensen, 1994; Bocquet, 2011; Iglesias et al., 2013) and extended Kalman filter (Jazwinski, 2007; Huang et al., 2022), in which fields are approximated using Gaussian kernels. However, these methods are hindered by computational complexity, which is characterized by time complexity and renders it infeasible for extremely large datasets(Angell & Sheldon, 2018; Yadav et al., 2021). There was also the introduction of model reduction techniques to manage high-dimensional fields and perform field reconstruction (Angell & Sheldon, 2018; Gu et al., 2017; Gherlone et al., 2012).

In deep learning, several techniques can be used for reconstruction, including super-resolution, neural inpainting, and implicit neural representation. Super-resolution techniques are primarily concerned with the enhancement of image quality from a lower to a higher resolution. However, within the domain of continuous field reconstruction of scientific datasets, the paired data for such processes are absent. Our principal objective lies in the construction of a continuous representation $u$ derived from a discretized data set. Recent advances in deep learning-driven super resolution facilitate continuous magnification through a variety of methodologies, including scale-consistent positional encodings (Ntavelis et al., 2022) and training with variable sizes (Chai et al., 2022). Furthermore, these methods encompass local conditioning strategies that take advantage of the deep features of the surrounding area (Chen et al., 2021) or employ neighborhood-based interpolation (Luo et al., 2023). Image inpainting techniques can be broadly classified into traditional and learning-based approaches. Traditional methods utilize low-level features through diffusion(Bertalmio et al., 2000) or patch-based (Barnes et al., 2009) techniques to propagate known information into missing regions. In contrast, learning-based approaches, including GANs(Yu et al., 2018; Lee et al., 2020) and diffusion models(Song et al., 2023b; Chung et al., 2022), produce more semantically coherent reconstructions but are often computationally intensive and may require additional post-processing.

**Implicit Neural Representation (INR):** INRs are a type of neural network that implicitly represents data (like images, 3D shapes, or other signals); that is, instead of storing data as arrays of values (like pixels in an image), INRs learn a continuous function that maps coordinates (like x,y,z) to the value at that point (like pixel brightness or density). Two primary purposes of INRs are (1) Parameterizing the sensor domain, i.e., INRs map sensor coordinates (like where a measurement was taken) to predicted sensor values; this is useful when real sensor data are noisy or incomplete, and the INR fills in or improves those data. e.g., MRI (Song et al., 2023a) sensors measure data in the Fourier domain. (2) Parameterizing the density domain, that is, mapping spatial coordinates (like a point in 2D or 3D space) to underlying physical density values, is a more direct representation of the actual object or field that is measured. e.g, Measurement of temperature from locations.

**Gabor filter in computer vision :** The application of Gabor filters for encoding spatial coordinates has proven to be a significant advancement for Implicit Neural Representations (INRs). For example, WIRE (Saragadam et al., 2023) uses a complex Gabor wavelet as activations to get richer location-dependent features. Multiplicative Filter Networks (MFN) Fathony et al. (2020) iteratively multiply together the linear functions of Gabor wavelet functions applied to the input to produce richer feature representations from input coordinates; the authors show this construction can be viewed as a linear function approximator operating over a large collection of Fourier/Gabor basis functions. The Multiplicative and Modulated Gabor Network (MMGN) (Luo et al., 2024) adopts an encoder-decoder architecture for location representations, and emphasizes that Gabor-based, localized frequency representations have an advantage over global Fourier bases when modeling complex, non-stationary patterns.

**Diffusion based Methods:** Recent generative approaches have advanced sparse-to-full reconstruction of physical dynamics. CoNFiLD Du et al. (2024) uses conditional neural-field latent diffusion to generate high-fidelity turbulent flows and support zero-shot reconstruction and restoration. SDIFT Chen et al. (2025) employs diffusion in Functional Tucker space with a message-passing posterior to recover full-field dynamics from irregular sparse measurements. S3GM Li et al. (2024) leverages an unconditional score-based generative prior and conditions it on sparse sensors, achieving robust and accurate reconstruction even under extreme sparsity and noise.In the training stage of these above methods, full-field data is necessary, which is often not feasible in many real-world scenarios. On the other hand, our goal is to learn the physical dynamics using only sparse data.

## C  ARCHITECTURE

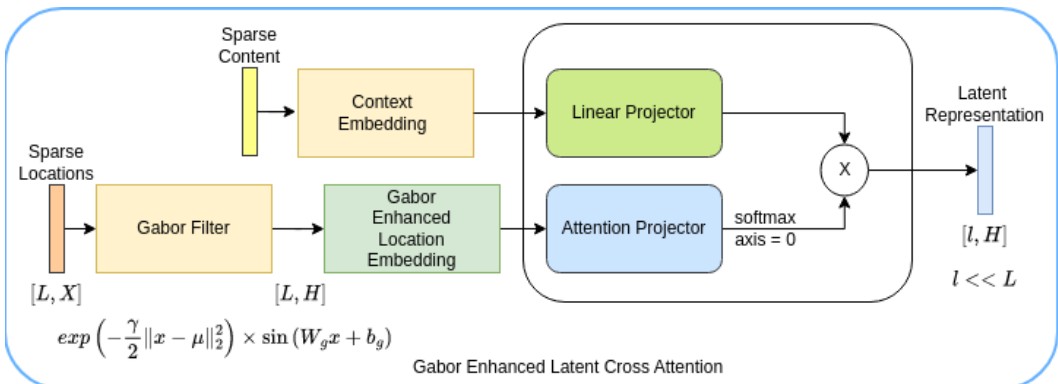

Figure 4: **Gabor Enhanced Latent Cross Attention Architecture**. The mechanism processes sparse spatial data through two parallel pathways. The upper pathway encodes sparse content through a Context Embedding module, which is then projected via an Linear Projector to linear projected features. The lower pathway transforms sparse locations [L, X] through a Gabor Filter—defined as the product of a Gaussian envelope and sinusoidal carrier: $\exp\left(-\frac{\gamma}{2}\|\mathbf{x} - \boldsymbol{\mu}\|_2^2\right) \times \sin(\mathbf{W}_g\mathbf{x} + \mathbf{b}_g))$ followed by a Gabor Enhanced Location Embedding to produce enriched positional representations [L, H]. These location embeddings pass through a Attention Projection (softmax along $axis = 0$), and the resulting attention weights are combined with the linear projected features via element-wise multiplication to produce a compact latent representation [l, H], where $l \ll L$. This cross-attention mechanism effectively compresses high-dimensional sparse inputs into a lower-dimensional latent space while preserving spatial frequency information through the Gabor filtering operation.

Figure 5 accompanies Figure 2 in the main paper. During training, only sparse locations are fed into the architecture. These sparse locations are processed through multiple Gabor filter layers with shared weights (bottom left). The sample sparse locations, but with the sparse content undergoes latent cross-attention processing (top left). The outputs feed into a Hierarchical KAN (Kolmogorov-Arnold Network) Block containing three sequential KAN modules. The hierarchical KAN outputs are then combined through element-wise multiplication operations and integrated with additional processing before passing through an MLP to generate the final sparse content output. During training, the model leverages sparse positional information rather than requiring dense location data, enabling its use for reconstruction tasks with sparse data.

## D  DATASET DETAILS

To test and validate the proposed methodology, we use four datasets that exhibit distinct characteristics. The first data set consists of the merged satellite product derived from a space-borne altimeter. The second data set consists of a data-assimilative forecast of a biological variable from a numerical ocean model. The third dataset is derived from a simulation of a prominent climate model. The fourth dataset consists of remotely sensed observations. All considered datasets are from the

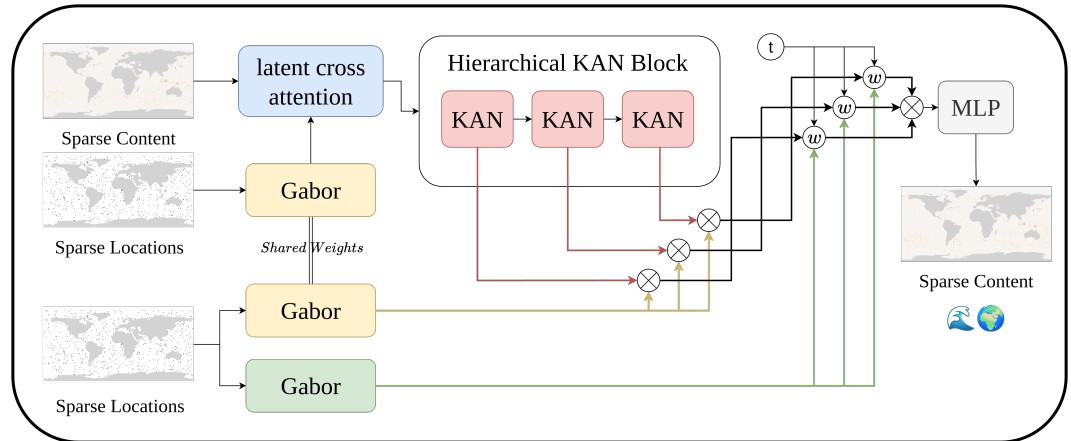

Figure 5: **KHINR Training architecture.** Sparse observation locations ($\mathbf{k}$) and content ($\mathbf{v}(\mathbf{k})$) are encoded through Gabor filters (Eq. 1) and latent cross-attention. Sparse query locations ($\mathbf{x}$) are similarly Gabor-encoded. Hierarchical KAN blocks (red) extract multi-scale features, which are fused with spatial and temporal information through multiplicative gating and an output MLP to produce reconstructed values $\hat{u}(\mathbf{x})$.

metocean domain and represent important aspects of the Earth with different spatiotemporal variability, distributional characteristics, and range. This selection allows us to assess the effectiveness of the proposed approach in reconstructing continuous fields from sparse observations using implicit neural representations (INRs) across various conditions and showcases its potential.

## D.1 SEA SURFACE HEIGHT

The first dataset we use comes from the Copernicus Marine Environment Monitoring Service (CMEMS) global gridded Level 4 Sea Surface Height product, processed by the DUACS (Data Unification and Altimeter Combination System) multimission altimeter data processing system. The product provides altimeter satellite gridded Sea Level Anomalies (SLA) computed with respect to a twenty-year [1993, 2012] mean, estimated using Optimal Interpolation that merges L3 along-track measurements from different altimeter missions available (Pujol, 2024).

The DUACS system processes data from all altimeter missions available, including Sentinel-6A, Jason-3, Sentinel-3A, Sentinel-3B, Saral/AltiKa, Cryosat-2, and HY-2B. All missions are homogenized with respect to a reference mission, and the processing is fitted to the global ocean. The system has been designed to ensure quality and stability of the sea level record, with calibration and validation activities funded by space agencies being essential to maintain the quality of these products.

The dataset represents a critical component for monitoring sea level rise and understanding climate evolution, with the system maintained and operated by CMEMS since May 2015, having previously been integrated into the CNES multi-mission ground segment SSALTO.

## D.2 CHLOROPHYLL CONCENTRATION

The second dataset we use comes from the Copernicus Marine Service (CMEMS) operational global biogeochemical analysis and forecast system, produced at Mercator-Ocean (Lamouroux et al., 2024). The system provides global chlorophyll mass concentration analysis and forecast products. We obtain and use the surface Chlorophyll-A concentration in a global grid of 1 degree spatial resolution.

The system is based on the NEMO (Nucleus for European Modeling of the Ocean) platform coupled with the PISCES (Pelagic Interaction Scheme of Carbon and Ecosystem Studies) biogeochemical model. PISCES is a biogeochemical model that simulates marine biological productivity and describes the biogeochemical cycles of carbon, oxygen, and the main nutrients (P, N, Si, Fe), rep-

resenting the marine biogeochemistry component of ocean modeling platforms and Earth System models. The coupled NEMO-PISCES system with data assimilation has been operational since April 2018, and in September 2019, Mercator Ocean International successfully updated its global biogeochemical analysis and forecasting system with Ocean Color data assimilation capability.

Satellite-derived surface chlorophyll data are assimilated weekly into the three-dimensional ensemble configuration of the coupled NEMO-PISCES model using a reduced-order Kalman filter, improving surface analysis and forecast chlorophyll representation across major parts of the model domain. The biogeochemical model is offline coupled with the dynamical ocean from the CMEMS global physical analysis and forecasting operational system, at a daily frequency, and benefits from the assimilation of satellite (SSH-SST-SIC) and in situ physical data. The time series is aggregated in time to reach a two full year's time series sliding window, with output mean fields interpolated on a standard regular grid in NetCDF format.

### D.3 GLOBAL SURFACE TEMPERATURE

The third dataset we use comes from the pre-industrial control run of the Community Earth System Model version 2 (CESM2). This dataset provides monthly averages of global surface temperature, representing a key atmospheric variable. The CESM2 dataset captures the complex atmospheric patterns, making it an ideal resource for evaluating our method.

CESM2 simulates interactions among major components of the Earth system, including the atmosphere, oceans, land surface, and sea ice. The pre-industrial control run specifically models a stable climate without human influence, serving as a reference point for understanding how the climate may change under future scenarios.

Because the data is simulated and exhibits rich spatial and temporal variation, it offers a robust environment for testing the performance of our proposed Implicit Neural Representation (INR) method. This dataset is part of the Coupled Model Intercomparison Project (CMIP) under the World Climate Research Program (WCRP), which supports standardized climate modeling experiments across research groups. The U.S. Department of Energy's Program for Climate Model Diagnosis and Intercomparison (PCMDI), in collaboration with the Global Organization for Earth System Science Portals, maintains and distributes this data through pcmdi.llnl.gov.

### D.4 SEA SURFACE TEMPERATURE

The fourth dataset we use comes from the Group for High Resolution Sea Surface Temperature (GHRSST) and is known as the Level 4 Multiscale Ultrahigh Resolution (MUR) Global Sea Surface Temperature (SST) dataset, which is available through NASA's Earthdata platform. It includes both retrospective data (available with a four-day delay) and near-real-time data (with a one-day delay), created using an optimal interpolation technique based on wavelet functions, mapped globally at a fine spatial resolution of 0.01 degrees. The version 4 MUR L4 analysis is based on nighttime GHRSST Level 2P skin and subskin SST observations from several instruments. The dataset incorporates observations from multiple sources, including the NASA AMSR-E, JAXA AMSR2 on GCOM-W1, MODIS on Aqua and Terra, U.S. Navy WindSat, NOAA AVHRR, and in situ SST from NOAA iQuam. Ice data are from the OSI SAF High Latitude Processing Center.

For this study, we focus on a subset of the dataset covering the western North Atlantic, specifically a region bounded by coordinates (34°N,-69°E) in the southwest and (44°N,-59°E) in the northeast. Unlike atmospheric data, oceanic fields exhibit unique spatial and temporal patterns. These range from small-scale features like eddies and thermal fronts to large-scale ocean currents and gyres. The GHRSST Level 4 MUR Global SST dataset captures this complexity through its ultra-high resolution, satellite-derived measurements.

This richness and level of detail make the dataset a strong benchmark for evaluating how well our method performs with real-world, high-resolution ocean data. We use one year of daily SST observations—from August 20, 2022, to August 20, 2023—focused on the Gulf Stream region. NASA provides this dataset freely as part of its Earth Science Data Systems (ESDS) Program, accessible through earthdata.nasa.gov.

# E  ADDITIONAL IMPLEMENTATION DETAILS

## E.1  MODEL ARCHITECTURE AND HYPERPARAMETERS

Model architecture design and hyperparameter optimization play a vital role in model performance, especially in the deep learning domain. To ensure a fair comparison between the architectures, each model was tuned on various ranges of hyperparameters. Table 6 contains the list of model configurations used to train the models. SIREN, FFN+P/G, MMGN, KHINR. Our best hyperparameters for different datasets are listed in Table 7 and 8.

Table 6: Hyperparameter configurations used for training different model architectures. Column 1 lists the various models used in this project. Column 2 specifies the hyperparameters and architectural variants associated with each model, and Column 3 depicts the range of values considered for each hyperparameter during training. These settings were used for model evaluation and optimization across the GST and SST datasets.

| Model | Hyperparameters | Values |
|---|---|---|
| SIREN | Width | $\{128, 192, 256, 384, 512\}$ |
|  | Depth | $\{3, 4, 5, 6, 7\}$ |
|  | Weight scale $w_0$ | $\{1, 5, 10, 15, 20, 25, 30, 35, 40, 45, 50, 100\}$ |
| FFN+P | Width | $\{128, 192, 256, 384, 512\}$ |
|  | Depth | $\{3, 4, 5, 6, 7\}$ |
|  | Frequency constant | $\{30, 40, 50, 60, 70, 80, 90, 100, 110, 120\}$ |
|  | Frequency number | $\{150, 160, 170, 180, 190, 200, 210, 220, 230, 240, 250\}$ |
|  | Non-linear activation | $\{$ReLU, Sigmoid, Tanh, SELU, GELU, Swish$\}$ |
| FFN+G | Width | $\{128, 192, 256, 384, 512\}$ |
|  | Depth | $\{3, 4, 5, 6, 7\}$ |
|  | Gaussian $\sigma$ | $\{1, 3, 5, 7, 10, 20, 30, 40, 50\}$ |
|  | Encode size | $\{64, 128, 256, 512\}$ |
|  | Non-linear activation | $\{$ReLU, Sigmoid, Tanh, SELU, GELU, Swish$\}$ |
| MMGN | Width | $\{128, 192, 256, 384, 512\}$ |
|  | Depth | $\{3, 4, 5, 6, 7\}$ |
|  | Input scale | $\{128, 256, 512\}$ |
|  | Latent size | $\{1, 2, 4, 8, 16, 32, 64, 128, 256, 512\}$ |
|  | Latent initialization | $\{$Uniform, Gaussian, Ones, Zeros, Orthogonal$\}$ |
| WIRE | Width | $\{128, 192, 256, 512\}$ |
|  | Depth | $\{2, 3, 4, 5, 6\}$ |
|  | Scale | $\{2, 4, 8, 10\}$ |
| KHINR | Latent dimension | $\{64, 128, 256, 512, 1024\}$ |
|  | No of Transformer Blocks | $\{1, 2, 3, 4, 5\}$ |
|  | MLP Layers | $\{1, 2, 3, 4, 5\}$ |
|  | Non-linear activation | $\{$ReLU, Sigmoid, Tanh, SELU, GELU, Swish$\}$ |
|  | Weight initialization | $\{$Uniform, Gaussian, Ones, Zeros, Orthogonal$\}$ |
|  | Alpha (Gabor) | $\{1/2, 1/4, 1/6, 1/\sqrt{2}, 1/\sqrt{4}, 1/\sqrt{6}\}$ |
|  | Beta (Gabor) | $\{1/2, 1/4, 1/6, 1/\sqrt{2}, 1/\sqrt{4}, 1/\sqrt{6}\}$ |

## E.2  BASELINE IMPLEMENTATION

SIREN (Sitzmann et al., 2020a) uses periodic activation functions with neural networks to represent signals and is ideally suited for representing complex natural signals and their derivatives.

FFN (Tancik et al., 2020a) introduced Fourier feature mappings to enable multilayer perceptrons (MLPs) to learn high-frequency functions in low dimensions, this has two variants with positional encoding (P) and Gaussian encoding (G).

MMGN (Luo et al., 2024) factorizes spatio-temporal variability into spatial and temporal components using the separation of variables technique and learns relevant basis functions from sparsely sampled irregular data points to represent represent continuous data. See Section 2.1 of the Supplementary Material for more details of the baselines.

WIRE (Saragadam et al., 2023) Inspired by Harmonic analysis, Wavelet Implicit Neural Representation (WIRE) uses complex gabor wavelet as its activation function that is well known to be optimally concentrated in space-frequency and effective biases for representing RGB images.

MFN (Fathony et al., 2020) The authors introduced multiplicative filter networks, namely GaborNet and FourierNet, as alternative functional representations of deep neural networks. These networks are constructed by sequentially applying sinusoidal or Gabor filter-based linear multiplicative functions. The authors demonstrate that, in certain scenarios, these proposed networks can surpass the performance of contemporary deep networks utilizing ReLu activations.

CMLP Ramasinghe & Lucey (2022a) shifts the focus in coordinate-MLPs away from periodic activations and positional embeddings towards a larger class of robust activations, enabling simpler, more powerful neural architectures for signal representation.

GridMix Wang et al. (2025) introduces a mixture of grid-based spatial modulations that capture global structure while preserving local detail, overcoming limitations of global and vanilla grid-based methods.

Senseiver Santos et al. (2022) is an attention-based framework that reconstructs high-dimensional spatial fields from extremely sparse and flexible sensor inputs by encoding them into a latent space via cross-attention and decoding efficiently at arbitrary resolutions.

INCODE Kazerouni et al. (2024) enhances implicit neural representations by dynamically controlling sinusoidal activations through a harmonizer–composer architecture that adapts to task-specific priors. This enables robust, high-fidelity reconstruction across audio, images, 3D shapes, NeRFs, and inverse problems such as denoising, super-resolution, and CT reconstruction.

FINER Liu et al. (2024a) introduces variable-periodic activation functions that flexibly tune an INR's supported frequency set by adjusting network biases, enabling better representation of complex multi-frequency signals. This leads to improved performance across 2D image fitting, 3D SDFs, and 5D NeRFs compared to existing INR methods.

In addition, We have also compared with Standard methods like DeepONet Lu et al. (2021), UNet Ronneberger et al. (2015) and Fourier Neural Operators Li et al. (2020).

All the models were trained on L2 loss for 200 epochs, using the learning rate of 0.001 with the weight decay of 0.0001 and AdamW optimizer. Batch sizes used were 16 and 2 due to variations in resolution between Earth simulation(GST) and satellite imagery(SST) data. All experiments were conducted using PyTorch Lightning on a single NVIDIA A6000 GPU with 48GB of memory. The implementation of KAN is inspired by (Yang & Wang, 2024).

Table 7: Best hyperparameters of KHINR on SSH and CHL datasets at all sampling ratios for different tasks.

| Model | SSH | | CHL | |
|---|---|---|---|---|
| | Task 1 & 2 | Task 3 & 4 | Task 1 & 2 | Task 3 & 4 |
| Num Blocks | 2 | 3 | 3 | 3 |
| Latent dim | 256 | 256 | 512 | 256 |
| Hidden dim | 256 | 256 | 512 | 256 |
| Num Layers (Embedding layers) | 3 | 2 | 4 | 2 |

### E.3 SAMPLING

Sampling for different tasks is applied differently for Task 1 (fixed observation locations and fixed number of points) and Task 3 (random observation locations and fixed number of points). Since the number of points is fixed, the sampling ratio is taken as specified. However for Task 2 (fixed

Table 8: Best hyperparameters of KHINR on GST and SST datasets at for different sampling ratios at all tasks.

| Model | GST | SST | |
|---|---|---|---|
| | 5,25,50 | 0.1 | 0.3, 0.5 |
| Num Blocks | 2 | 2 | 3 |
| Latent dim | 256 | 64 | 256 |
| Hidden dim | 256 | 128 | 256 |
| Num Layers (Embedding layers) | 4 | 4 | 2 |

Table 9: Minimum and maximum sampling ratios used during training for all the datasets. These sampled input locations are used to train the model for continuous field reconstruction from sparse observations.

| Data | GST | | | SST | | | SSH | | | CHL | | |
|---|---|---|---|---|---|---|---|---|---|---|---|---|
| Ratios | 5% | 25% | 50% | 0.10% | 0.30% | 0.50% | 10% | 25% | 50% | 10% | 25% | 50% |
| Min | 1 | 1 | 1 | 0.01 | 0.01 | 0.01 | 5 | 5 | 5 | 5 | 5 | 5 |
| Max | 5 | 25 | 50 | 0.10 | 0.30 | 0.50 | 10 | 25 | 50 | 10 | 25 | 50 |

observation locations and random number of points and Task 4 (random observation locations and random number of points) since the number of data points varies, in that case we give a range of sampling values as indicated in Table 9 to ensure fair evaluation of the model performance.

### E.4 TASKS

To test how well our model handles different conditions, we designed a series of experiments that gradually increase in complexity. These include: *Task 1*, where both the number and location of input points are fixed; *Task 2*, which uses a random number of points but keeps their locations fixed; *Task 3*, which fixes the number of points but places them at random locations; and *Task 4*, where both the number and locations of points are randomized.

## F    ADDITIONAL MODEL ANALYSIS

### F.1    COMPUTATIONAL ANALYSIS

Figure 7 illustrates the trade-off between inference time and model ranking for various implicit neural representation (INR) models, where the circle size corresponds to the number of parameters. Models such as SIREN and MMGN, despite having larger parameter counts, exhibit significantly higher inference times, making them less efficient. On the other hand, lightweight models like WIRE and MFN achieve faster inference but at the cost of lower ranking. KHINR (ours), with a moderate parameter count of 0.4M, achieves the best overall ranking while maintaining a low inference time comparable to smaller models. This demonstrates KHINR's superior balance of efficiency and performance, outperforming existing baselines by offering both speed and representational power.

### F.2    STATISTICAL ANALYSIS

Table 10 summarizes the quantitative performance of KHINR over five independent runs for SST, GST, and SSH. The results show that across five independent runs, KHINR exhibits very low run-to-run variability, indicating strong stability and reproducibility. The standard deviations for all metrics—particularly SSIM (on the order of $10^{-4}$–$10^{-3}$) and PSNR ($\leq 0.05$ dB)—are negligible, implying that performance fluctuations due to random initialization or training nondeterminism are minimal. The mean MSE values closely match those reported in Tables 1 and 2, further corroborating the model's robustness. Overall, these results show that KHINR delivers reliable, repeatable predictions across trials.

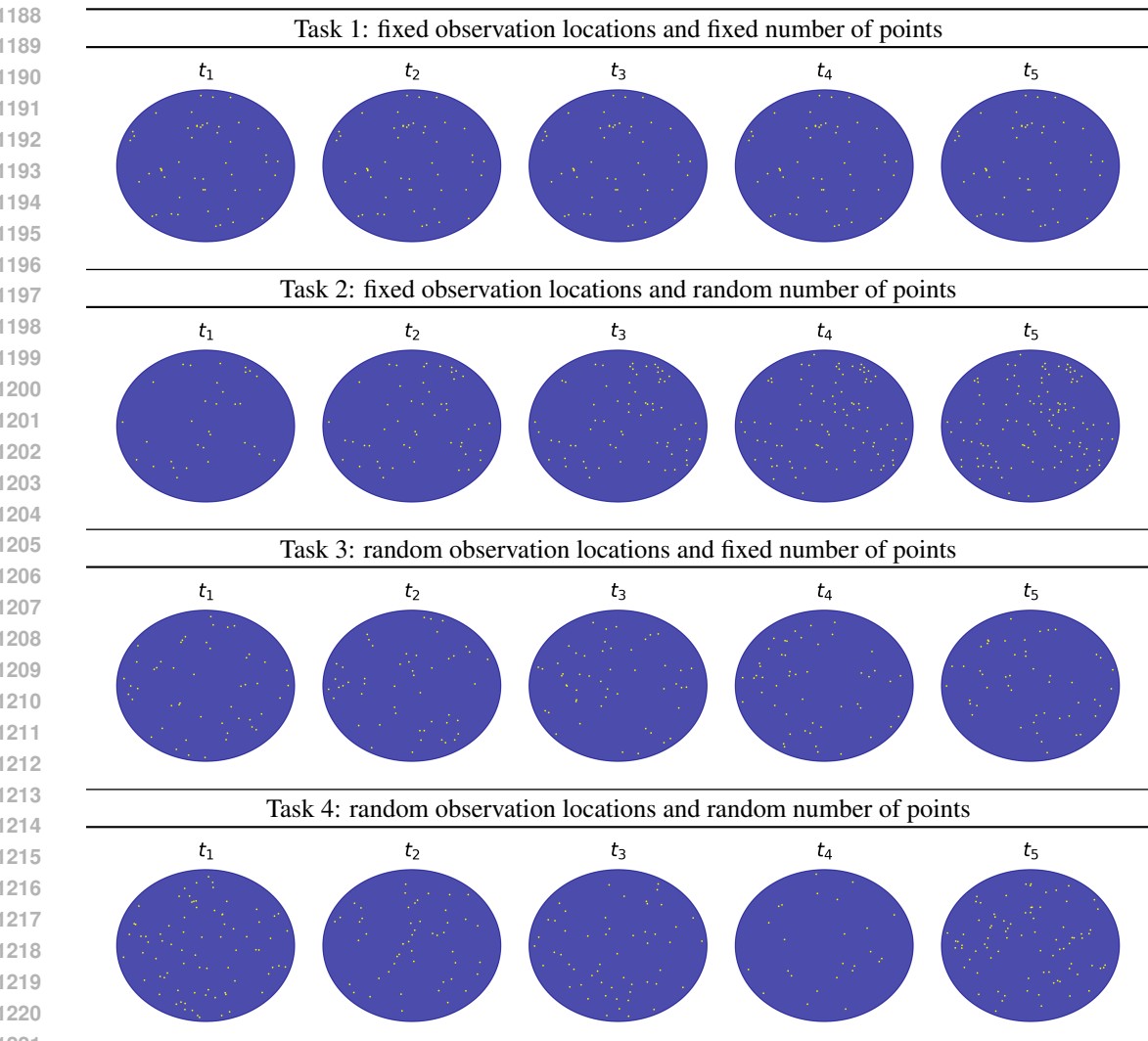

Figure 6: Diagram illustrating different field reconstruction tasks arranged in order of increasing complexity for model evaluation. Each task differs based on the degree of randomness in the number and spatial distribution of data points. These tasks allow the model evaluation over varying sampling conditions.

Table 10: Quantitative Metrics of KHINR obtained on five runs.

|  | SST | | | GST | | | SSH | | |
| --- | --- | --- | --- | --- | --- | --- | --- | --- | --- |
|  | MSE($10^{-3}$) | PSNR | SSIM | MSE($10^{-3}$) | PSNR | SSIM | MSE($10^{-3}$) | PSNR | SSIM |
| 0 | 0.81 | 31.14 | 0.9175 | 3.52 | 24.54 | 0.7319 | 1.22 | 29.68 | 0.9230 |
| 1 | 0.81 | 31.12 | 0.9168 | 3.55 | 24.49 | 0.7324 | 1.22 | 29.68 | 0.9232 |
| 2 | 0.82 | 31.11 | 0.9179 | 3.50 | 24.56 | 0.7336 | 1.21 | 29.71 | 0.9240 |
| 3 | 0.79 | 31.23 | 0.9176 | 3.50 | 24.56 | 0.7328 | 1.22 | 29.69 | 0.9231 |
| 4 | 0.81 | 31.15 | 0.9186 | 3.53 | 24.52 | 0.7328 | 1.22 | 29.69 | 0.9230 |
| mean | 0.808 | 31.15 | 0.9177 | 3.52 | 24.53 | 0.7327 | 1.218 | 29.69 | 0.9233 |
| std deviation | 0.0098 | 0.0424 | 0.0006 | 0.0190 | 0.0265 | 0.0002 | 0.0040 | 0.0109 | 0.0004 |

### F.3  GABOR ENHANCED LATENT ATTENTION

Table 11 demonstrates that KHINR achieves consistently lower MSE while also reducing training time 3 when equipped with the Gabor-enhanced latent attention. This result indicates that the Gabor-enhanced latent representation improves the model's ability to capture relevant features more

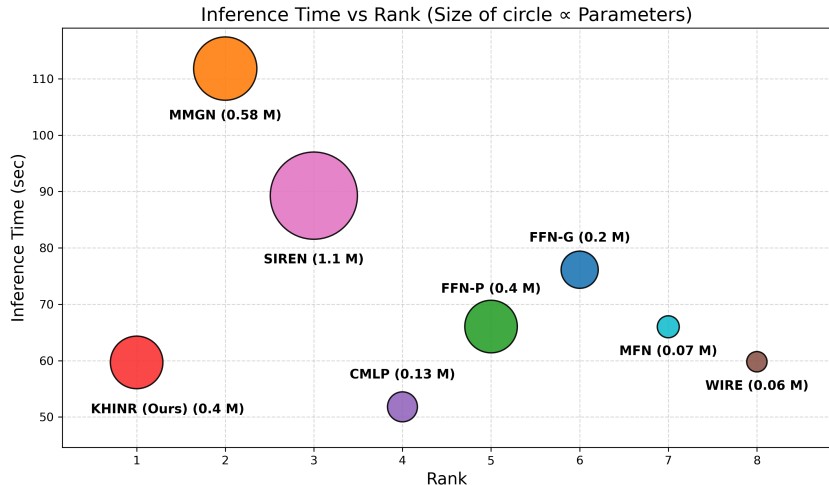

Figure 7: Inference time vs. model ranking plot for various INR models, with circle size indicating parameter count. KHINR achieves the best ranking with the comparable inference time and similar model size, demonstrating superior efficiency and performance compared to other baselines for SST dataset Task 1 with 0.1% sampling ratio.

Table 11: Comparison between KHINR with and without Gabor enhanced latent attention mechanism for the SSH and SST datasets with sampling ratios of 10% and 0.1%, respectively.

|  | Dataset | SSH | SSH | SST | SST |
|---|---|---|---|---|---|
|  |  | Task1 | Task3 | Task1 | Task3 |
| w/o | MSE($10^{-3}$) | 4.48 | 1.30 | 0.88 | 0.64 |
|  | PSNR | 23.49 | 28.87 | 30.67 | 32.14 |
|  | SSIM | 0.7945 | 0.9195 | 0.9132 | 0.9195 |
| w | MSE($10^{-3}$) | **4.24** | **1.03** | **0.87** | **0.49** |
|  | PSNR | **23.73** | **29.89** | **30.69** | **33.19** |
|  | SSIM | **0.7999** | **0.9341** | **0.9135** | **0.9252** |

efficiently than the architecture excluding Gabor-enhanced latent attention with less computational time.

## F.4 SENSITIVITY STUDIES

### F.4.1 LATENT DIMENSION

Table 12: Sensitivity study of the latent dimension on GST and SST datasets at sampling ratios of 5% and 0.3%, respectively, for Tasks 1 and 4, evaluated using the MSE($10^{-3}$) metric

| Dimension | GST (Task 1) | GST (Task 4) | SST (Task 1) | SST (Task 4) |
|---|---|---|---|---|
| 32 | 4.21 | 3.51 | 0.36 | 0.39 |
| 64 | 3.90 | **3.50** | 0.34 | 0.38 |
| 128 | 3.84 | 3.52 | **0.33** | 0.39 |
| 256 | **3.69** | **3.50** | 0.34 | **0.37** |

Table 12 presents the sensitivity of KHINR to the latent dimensionality (32, 64, 128, 256) for GST and SST with sampling ratios of 5% and 0.3% for Tasks 1 and 4. Increasing the latent dimension steadily reduces the GST error (Task 1 decreases from 4.21 to 3.69), underscoring the increased representational capacity afforded by larger latent spaces. For SST, the lowest or near-lowest errors

occur at a latent dimension of 128, indicating a point of saturation where further increases provide only marginal benefit. These results confirm that the latent dimension selected in our model achieves an effective trade-off between parameter count and reconstruction error, balancing expressivity with efficiency.

### F.4.2   HIDDEN DIMENSION

Table 13: Sensitivity study of number of hidden dimension for GST and SST with sampling ratios of 5% and 0.3% respectively for Tasks 1 and 4, evaluated using the MSE($10^-3$) metric

| Dimension | GST (Task 1) | GST (Task 4) | SST (Task 1) | SST (Task 4) |
|---|---|---|---|---|
| 32 | 10.41 | 10.30 | 1.41 | 1.64 |
| 64 | 6.84 | 6.81 | 0.75 | 0.84 |
| 128 | **3.69** | 4.55 | 0.56 | 0.51 |
| 256 | 3.91 | **3.50** | **0.34** | **0.37** |

Table 13 tabulates the sensitivity of KHINR to the number of hidden dimensions (32, 64, 128, 256) for GST and SST with sampling ratios of 5% and 0.3% for Tasks 1 and 4. Increasing the hidden dimension dramatically reduces the GST error from 10.41 at 32 to 3.69 at 128 for Task 1, and from 10.30 to 3.50 for Task 4, highlighting the strong impact of hidden capacity on model expressivity. For SST, errors likewise drop steeply (from 1.41 at 32 to 0.34 at 256 for Task 1 and from 1.64 to 0.37 for Task 4), indicating that larger hidden spaces enable more accurate reconstructions. The improvement plateaus beyond 128 for GST and around 256 for SST, suggesting that the chosen hidden dimension provides a good trade-off between parameter count and performance.

### F.4.3   HIERARCHAL BLOCKS

Table 14: Sensitivity study of number of hierarchical blocks for GST and SST with sampling ratios of 5% and 0.3% respectively for Tasks 1 and 4, evaluated using the MSE($10^-3$) metric

| Number of Blocks | GST (Task 1) | GST (Task 4) | SST (Task 1) | SST (Task 4) |
|---|---|---|---|---|
| 1 | 4.56 | 4.30 | 0.46 | 0.63 |
| 2 | 3.69 | 3.50 | 0.36 | 0.43 |
| 3 | 4.03 | **3.32** | 0.34 | **0.37** |
| 4 | **3.45** | 3.33 | **0.32** | 0.39 |

Table 14 represents sensitivity of KHINR to the number of heirarchical blocks (1,2,3,4) for GST and SST with sampling ratios of 5% and 0.3% for Tasks 1 and 4. The results, as detailed in Table 14, demonstrate a clear relationship between the number of blocks and the model's performance. As shown in the table, increasing the number of blocks from one to two yields a significant performance improvement across all evaluated scenarios. This suggests that a deeper architecture is crucial for capturing the complex hierarchical dependencies within the data. For Task 1, the optimal performance is achieved with four blocks for both GST (3.45) and SST (0.32). In contrast, for Task 4, the best results are obtained with three hierarchical blocks for both GST (3.32) and SST (0.37). Based on this analysis, a trade-off between model depth and task-specific performance is observed. While a deeper model with four blocks is advantageous for Task 1, a slightly shallower architecture with three blocks is optimal for Task 4. This highlights the importance of tuning model depth to the specific characteristics of the task and dataset.

## G   ADDITIONAL RESULTS

### G.1   QUANTITATIVE RESULTS

### G.1.1 NON-INR BASELINES

We evaluated our proposed KHINR model against three widely used architectures: UNet(Ronneberger et al., 2015), FNO(Li et al., 2020), and DeepONet(Lu et al., 2021) on both the GST and SST datasets across multiple sampling ratios. As shown in Table 15, KHINR consistently achieves the lowest MSE across all tasks and sampling settings. Under extremely sparse observations (5% sampling for GST and 0.1% for SST), KHINR outperforms all baselines with a significant margin, achieving 3.69/2.56 on GST (Tasks 1 and 3) and 2.01/1.92 on SST. In contrast, Non-INR baselines show substantially higher errors, with UNet and FNO performing an order of magnitude worse in several cases. Even when the sampling ratio is increased (25% for GST and 0.3% for SST), KHINR maintains its advantage, achieving 0.87/0.49 on GST and 0.34/0.21 on SST, again outperforming all baselines by a clear margin. These results demonstrate that KHINR is highly effective at reconstructing fields under both extremely sparse and moderately sparse sampling scenarios, while conventional Non-INR architectures struggle with the same level of sparsity. The consistent performance gain illustrates the robustness and generalization capacity of KHINR compared to existing reconstruction models.

Table 15: Comparison of KHINR with other Non-INR baselines on GST and SST datasets at different sampling ratios for different tasks using MSE($10^{-3}$) metrics. Best results are in **bold**.

| Model | GST | | SST | |
|---|---|---|---|---|
| | Task 1 | Task 3 | Task 1 | Task 3 |
| *Sampling ratio = 5% (GST), 0.1% (SST)* | | | | |
| KHINR | **3.69** | **2.56** | **2.01** | **1.92** |
| UNet | 23.32 | 20.35 | 20.25 | 19.93 |
| FNO | 20.45 | 19.65 | 18.73 | 19.60 |
| DeepONet | 21.27 | 21.19 | 20.80 | 21.09 |
| *Sampling ratio = 25% (GST), 0.3% (SST)* | | | | |
| KHINR | **0.87** | **0.49** | **0.34** | **0.21** |
| UNet | 78.2 | 83.5 | 65.9 | 67.14 |
| FNO | 2.84 | 2.86 | 2.34 | 2.18 |
| DeepONet | 8.22 | 7.86 | 8.08 | 8.43 |

### G.1.2 ADDITIONAL BASELINES

Table 16 reports extended baseline comparisons on the GST and SSH datasets across four reconstruction tasks under two sampling regimes. At extremely sparse sensing (5% for GST and 10% for SSH), KHINR consistently achieves the lowest MSE across all tasks, outperforming INCODE Kazerouni et al. (2024), FINER Liu et al. (2024a), and SENSEIVER Santos et al. (2022) by a significant margin. This performance gap persists even at higher sampling ratios (25% for GST and 25% for SSH), where KHINR again provides the most accurate reconstructions. These results demonstrate the robustness of our method in both low and moderately-sampled settings and highlight its capability across diverse tasks. The poor performance of SENSEIVER is attributed to its grid-like reconstruction patterns, as illustrated in Figure 2 of Santos et al. (2022).

### G.1.3 EXTENDED RESULTS

To evaluate the performance of our proposed model quantitatively, we conducted a comprehensive quantitative analysis using the Peak Signal-to-Noise Ratio (PSNR) and the Structural Similarity Index Measure (SSIM) as our primary metrics. These metrics are standard benchmarks for assessing image reconstruction quality, where higher values for both PSNR and SSIM indicate a higher fidelity and perceptual quality of the generated result compared to the ground truth. The detailed results of our comparison are presented in Tables 17, 18, 19, and 20. These tables document the performance of our model, KHINR, against several leading baseline models (SIREN, FFN-P, FFN-G, MMGN, CMLP, MFN, and WIRE) across four distinct datasets, multiple tasks, and a wide range of data sampling ratios.

Table 16: Comparison of KHINR with other INR baselines on GST and SSH datasets at different sampling ratios for different tasks using MSE($10^{-3}$) metrics. Best results are in **bold**.

| Model | GST | | | | SSH | | | |
|---|---|---|---|---|---|---|---|---|
| | Task 1 | Task 2 | Task 3 | Task 4 | Task 1 | Task 2 | Task 3 | Task 4 |
| *Sampling ratio = 5% (GST), 10% (SSH)* | | | | | | | | |
| KHINR | **3.69** | **4.44** | **2.56** | **3.50** | **4.24** | 4.29 | **1.03** | **1.46** |
| INCODE | 18.11 | 19.24 | 15.05 | 15.09 | 5.82 | **3.25** | 2.02 | 2.05 |
| FINER | 18.89 | 20.43 | 14.97 | 14.99 | 5.12 | 5.25 | 1.93 | 1.95 |
| SENSEIVER | 33.04 | 32.29 | 9.09 | 8.84 | 8.70 | 8.87 | 4.94 | 6.53 |
| GridMix | 201.04 | 192.25 | 80.35 | 78.65 | 76.25 | 81.22 | 42.34 | 53.96 |
| *Sampling ratio = 25% (GST), 25% (SSH)* | | | | | | | | |
| KHINR | **2.01** | **2.41** | **1.92** | **2.27** | **2.89** | **2.89** | **0.46** | **0.72** |
| INCODE | 13.98 | 14.50 | 13.59 | 13.67 | 5.79 | 3.72 | 1.67 | 1.62 |
| FINER | 13.73 | 13.99 | 13.14 | 13.37 | 3.30 | 3.56 | 1.59 | 1.62 |
| SENSEIVER | 42.50 | 18.40 | 7.09 | 17.16 | 6.78 | 6.58 | 5.09 | 5.31 |
| GridMix | 180.56 | 170.42 | 102.59 | 101.26 | 60.45 | 59.54 | 22.65 | 22.89 |

A consistent and prominent trend across all evaluated scenarios is the superior performance of our proposed KHINR model. In the vast majority of cases, KHINR achieves state-of-the-art (SOTA) results, recording the highest PSNR and SSIM scores. This is particularly evident in low sampling ratios s.a. $0.1\%$ and $0.3\%$ on the SST dataset (Table 18), where KHINR establishes a significant performance margin over all other baselines. For instance, in Task 3 on the SST dataset with $0.5\%$ sampling, KHINR achieves a PSNR of 37.42, substantially outperforming the next best model, MMGN (34.56). This demonstrates our model's robust capability to reconstruct high-quality signals from very sparse inputs.

It is expected, all models exhibit improved performance as the sampling ratio increases, providing more data for the reconstruction task. However, it is notable that KHINR's performance scales more effectively than its competitors. For example, on the CHL dataset (Table 20), as the sampling ratio increases from $10\%$ to $50\%$, KHINR consistently widens its lead over the second-best performing model, FFN-P. In Task 1 at $10\%$ sampling, KHINR leads with a PSNR of 18.40 against FFN-P's 16.45, and this gap is maintained at the $50\%$ sampling ratio (22.83 for KHINR vs. 20.92 for FFN-P). This indicates that our model is not only effective in low-data scenarios but is also more proficient at leveraging additional information to further enhance reconstruction quality.

## G.2 QUALITATIVE RESULTS

In Figure 8 and 9 plots using different models for different datasets is depicted. Beyond the quantitative metrics, a qualitative inspection of the visual results provides crucial insight into the performance of each model. The reconstructed images, presented in Figure 8, 9, 10, 11, 12, and 13, visually corroborate the superiority of our proposed KHINR model. It is noticed that KHINR model is able to generate significantly sharper, clearer, and more structurally accurate results. The Wire and Siren baselines were only predicting the mean, which is why we did not include their visualizations.

The evaluation on the SSH and CHL datasets (Figure 8, 10, 12) further highlights KHINR's strengths. The CHL dataset features distinct, filament-like chlorophyll structures. These intricate patterns are crucial as they trace the underlying ocean currents and eddies, and our model's ability to capture them demonstrates a high degree of spatial fidelity, closely mirroring the reference plot. Other models tend to blur these critical features into indistinct shapes. Similarly, for the sparse SSH data, KHINR accurately reproduces the sharp, localized nature of the signals. In comparison, other baselines often introduce blurry artifacts or fail to resolve the sparse features correctly, again demonstrating the superiority of our model in capturing the true underlying structure of the data.

As illustrated in Figure 9,11, 13, the GST and SST datasets are characterized by complex, fine-scale textural patterns. Our model, KHINR, excels at capturing these high-frequency details, preserving the sharp boundaries and intricate swirls present in the reference images. In contrast, competing

Table 17: Comparison of KHINR and various INR baselines using PSNR and SSIM metrics on the GST Dataset, evaluated across diverse sampling ratios and multiple tasks.

| Task | Metric | SIREN | FFN-P | FFN-G | MMGN | CMLP | MFN | WIRE | KHINR |
|------|--------|-------|-------|-------|------|------|-----|------|-------|
| | | | | 5% sampling | | | | | |
| Task 1 | PSNR | 18.14 | 20.28 | 17.44 | 23.42 | 16.98 | 17.12 | 16.15 | **24.33** |
| | SSIM | 0.4853 | 0.4833 | 0.3880 | 0.6899 | 0.4121 | 0.3695 | 0.3813 | **0.6963** |
| Task 2 | PSNR | 18.01 | 19.93 | 17.07 | 23.01 | 16.86 | 16.83 | 16.15 | **23.53** |
| | SSIM | 0.4676 | 0.4556 | 0.3505 | **0.6844** | 0.4024 | 0.3449 | 0.3809 | 0.6743 |
| Task 3 | PSNR | 18.66 | 21.15 | 18.51 | 25.14 | 17.12 | 18.01 | 16.15 | **25.91** |
| | SSIM | 0.5650 | 0.6096 | 0.5162 | **0.7725** | 0.5064 | 0.4913 | 0.3817 | 0.7602 |
| Task 4 | PSNR | 18.66 | 21.19 | 18.51 | 24.22 | 17.25 | 18.01 | 16.15 | **24.55** |
| | SSIM | 0.5645 | 0.6066 | 0.5160 | **0.7564** | 0.5021 | 0.4913 | 0.3815 | 0.7332 |
| | | | | 25% sampling | | | | | |
| Task 1 | PSNR | 19.12 | 21.66 | 18.67 | 25.28 | 17.25 | 18.11 | 16.15 | **26.95** |
| | SSIM | 0.6139 | 0.6209 | 0.5487 | 0.7649 | 0.4970 | 0.5042 | 0.3822 | **0.7767** |
| Task 2 | PSNR | 5.65 | 21.46 | 18.71 | 24.96 | 17.35 | 18.01 | 16.15 | **26.18** |
| | SSIM | 0.0006 | 0.5924 | 0.5329 | 0.7538 | 0.4943 | 0.4852 | 0.3695 | **0.7585** |
| Task 3 | PSNR | 19.08 | 21.80 | 18.88 | 25.62 | 17.37 | 18.22 | 16.15 | **27.17** |
| | SSIM | 0.6203 | 0.6387 | 0.5565 | 0.7831 | 0.5165 | 0.5262 | 0.3822 | **0.7851** |
| Task 4 | PSNR | 19.06 | 21.86 | 18.78 | 25.35 | 17.35 | 18.19 | 16.13 | **26.44** |
| | SSIM | 0.6218 | 0.6444 | 0.5564 | **0.7777** | 0.5124 | 0.5241 | 0.3581 | 0.7693 |
| | | | | 50% sampling | | | | | |
| Task 1 | PSNR | 19.12 | 21.73 | 18.85 | 25.58 | 17.35 | 18.20 | 16.15 | **27.18** |
| | SSIM | 0.6238 | 0.6430 | 0.5653 | 0.7772 | 0.5153 | 0.5277 | 0.3823 | **0.7857** |
| Task 2 | PSNR | 19.10 | 21.69 | 18.87 | 25.35 | 17.39 | 18.18 | 16.14 | **26.63** |
| | SSIM | 0.6456 | 0.6189 | 0.5602 | **0.7703** | 0.4919 | 0.5156 | 0.36823 | 0.7698 |
| Task 3 | PSNR | 19.20 | 21.87 | 18.97 | 25.67 | 17.48 | 18.21 | 16.15 | **27.28** |
| | SSIM | 0.6357 | 0.6429 | 0.5694 | 0.7844 | 0.5173 | 0.5291 | 0.3823 | **0.7871** |
| Task 4 | PSNR | 19.14 | 21.83 | 18.86 | 25.49 | 17.55 | 18.21 | 16.16 | **26.74** |
| | SSIM | 0.6323 | 0.6435 | 0.5599 | **0.7808** | 0.5157 | 0.5254 | 0.3723 | 0.7753 |

models such as MFN and FFN produce results that are unable to capture the intensity, losing the essential structural information. While MMGN performs reasonably well, it fails to match the quality, spatial clarity and sharpness of KHINR. The output from CMLP is particularly blurry, failing almost entirely to reconstruct the underlying complex structures. This demonstrates KHINR's superior ability to preserve fine details, which is a key factor in its high SSIM scores.

Table 18: Comparison of KHINR and various INR baselines using PSNR and SSIM metrics on the SST Dataset, evaluated across diverse sampling ratios and multiple tasks.

| Task | Metric | SIREN | FFN-P | FFN-G | MMGN | CMLP | MFN | WIRE | KHINR |
|---|---|---|---|---|---|---|---|---|---|
| | | | | | 0.1% sampling | | | | |
| Task 1 | PSNR | 27.32 | 24.36 | 19.05 | 28.39 | 25.42 | 18.05 | 10.07 | **30.69** |
| | SSIM | 0.9038 | 0.6640 | 0.7528 | 0.9087 | 0.8914 | 0.7767 | 0.6317 | **0.9132** |
| Task 2 | PSNR | 26.39 | 23.75 | 18.67 | 27.43 | 25.05 | 18.24 | 10.09 | **29.63** |
| | SSIM | 0.8999 | 0.6482 | 0.7751 | 0.9050 | 0.8916 | 0.7563 | 0.6156 | **0.9056** |
| Task 3 | PSNR | 29.81 | 29.83 | 28.03 | 31.65 | 26.28 | 26.55 | 10.07 | **33.19** |
| | SSIM | 0.9209 | 0.8537 | 0.8905 | **0.9330** | 0.9023 | 0.8834 | 0.6318 | 0.9252 |
| Task 4 | PSNR | 29.20 | 29.08 | 26.37 | 28.45 | 26.58 | 26.11 | 10.07 | **31.17** |
| | SSIM | 0.9168 | 0.8308 | 0.8634 | 0.9089 | 0.9029 | 0.8767 | 0.6027 | **0.9170** |
| | | | | | 0.3% sampling | | | | |
| Task 1 | PSNR | 30.88 | 29.63 | 27.91 | 32.32 | 26.64 | 24.32 | 10.07 | **34.78** |
| | SSIM | 0.9264 | 0.8381 | 0.9083 | 0.9307 | 0.9036 | 0.8664 | 0.6266 | **0.9305** |
| Task 2 | PSNR | 29.55 | 28.47 | 24.62 | 31.14 | 26.45 | 22.88 | 10.63 | **32.96** |
| | SSIM | 0.9574 | 0.8017 | 0.8390 | **0.9264** | 0.9013 | 0.8462 | 0.5699 | 0.9240 |
| Task 3 | PSNR | 32.83 | 33.07 | 31.45 | 31.14 | 26.84 | 28.29 | 10.07 | **36.88** |
| | SSIM | 0.9395 | 0.9242 | 0.9292 | **0.9446** | 0.9056 | 0.9068 | 0.6666 | 0.9424 |
| Task 4 | PSNR | 32.25 | 32.57 | 28.92 | 32.24 | 26.65 | 27.61 | 10.36 | **34.70** |
| | SSIM | **0.9372** | 0.9165 | 0.8998 | 0.9367 | 0.9043 | 0.8976 | 0.6196 | 0.9345 |
| | | | | | 0.5% sampling | | | | |
| Task 1 | PSNR | 32.73 | 31.63 | 30.01 | 33.60 | 27.49 | 26.65 | 10.07 | **36.45** |
| | SSIM | 0.9378 | 0.8882 | 0.9168 | 0.9372 | 0.9077 | 0.8969 | 0.6467 | **0.9379** |
| Task 2 | PSNR | 31.71 | 30.66 | 28.86 | 32.45 | 26.93 | 25.71 | 13.64 | **34.52** |
| | SSIM | 0.9308 | 0.8659 | 0.9074 | **0.9333** | 0.9046 | 0.8740 | 0.5684 | 0.9314 |
| Task 3 | PSNR | 34.04 | 33.73 | 30.97 | 34.56 | 27.05 | 28.12 | 10.22 | **37.42** |
| | SSIM | 0.9456 | 0.9326 | 0.9235 | **0.9457** | 0.9066 | 0.9043 | 0.5547 | 0.9445 |
| Task 4 | PSNR | 5.39 | 33.44 | 30.79 | 33.16 | 27.06 | 28.21 | 11.08 | **35.44** |
| | SSIM | 0.0077 | 0.9288 | 0.9264 | **0.9413** | 0.9059 | 0.9037 | 0.5632 | 0.9380 |

Table 19: Comparison of KHINR and various INR baselines using PSNR and SSIM metrics on the SSH Dataset, evaluated across diverse sampling ratios and multiple tasks.

| Task | Metric | SIREN | FFN-P | FFN-G | MMGN | CMLP | MFN | WIRE | KHINR |
|---|---|---|---|---|---|---|---|---|---|
| | | | | 10% sampling | | | | | |
| Task 1 | PSNR | 24.51 | **29.01** | 28.40 | 24.02 | 28.96 | 27.87 | 27.34 | 23.73 |
| | SSIM | 0.0998 | 0.7343 | 0.7325 | 0.7280 | 0.7599 | 0.7324 | 0.6573 | **0.7909** |
| Task 2 | PSNR | 24.51 | **29.05** | 28.52 | 24.98 | 28.85 | 27.96 | 27.34 | 23.67 |
| | SSIM | 0.0998 | 0.7342 | 0.7348 | 0.7169 | 0.7472 | 0.7098 | 0.6573 | **0.7913** |
| Task 3 | PSNR | 24.51 | **35.82** | 33.55 | 31.58 | 31.53 | 31.59 | 27.34 | 29.89 |
| | SSIM | 0.0998 | 0.9207 | 0.8754 | 0.8648 | 0.8326 | 0.8247 | 0.6574 | **0.9341** |
| Task 4 | PSNR | 24.51 | **35.77** | 33.26 | 30.55 | 31.44 | 31.52 | 27.34 | 28.36 |
| | SSIM | 0.0998 | **0.9190** | 0.8679 | 0.8459 | 0.8305 | 0.8214 | 0.6574 | 0.9116 |
| | | | | 25% sampling | | | | | |
| Task 1 | PSNR | 24.51 | 30.42 | **30.94** | 28.40 | 30.38 | 30.65 | 27.35 | 25.39 |
| | SSIM | 0.0998 | 0.8123 | 0.8357 | 0.8077 | 0.8056 | 0.8016 | 0.6576 | **0.8532** |
| Task 2 | PSNR | 24.51 | 30.45 | **30.69** | 28.20 | 30.11 | 30.22 | 27.40 | 25.39 |
| | SSIM | 0.0998 | 0.8026 | 0.8152 | 0.8010 | 0.7824 | 0.7831 | 0.6596 | **0.8512** |
| Task 3 | PSNR | 24.51 | **37.43** | 35.38 | 33.55 | 29.86 | 31.82 | 27.35 | 33.35 |
| | SSIM | 0.0998 | 0.9361 | 0.9032 | 0.8962 | 0.7583 | 0.8321 | 0.6576 | **0.9627** |
| Task 4 | PSNR | 24.51 | **37.24** | 35.04 | 32.46 | 29.71 | 31.88 | 27.32 | 31.45 |
| | SSIM | 0.0998 | 0.9341 | 0.8972 | 0.8779 | 0.7537 | 0.8345 | 0.6545 | **0.9498** |
| | | | | 50% sampling | | | | | |
| Task 1 | PSNR | 24.51 | 33.20 | **33.75** | 33.04 | 31.70 | 31.66 | 27.35 | 28.57 |
| | SSIM | 0.0998 | 0.8951 | 0.8855 | 0.8886 | 0.8410 | 0.8284 | 0.6578 | **0.9232** |
| Task 2 | PSNR | 24.52 | 32.77 | **33.20** | 31.70 | 31.14 | 31.25 | 27.49 | 28.04 |
| | SSIM | 0.0991 | 0.8790 | 0.8731 | 0.8601 | 0.8211 | 0.8169 | 0.6633 | **0.9162** |
| Task 3 | PSNR | 24.51 | **38.03** | 36.01 | 34.25 | 32.37 | 32.07 | 27.35 | 34.60 |
| | SSIM | 0.0998 | 0.9419 | 0.9117 | 0.9086 | 0.8540 | 0.8391 | 0.6578 | **0.9686** |
| Task 4 | PSNR | 24.62 | **37.86** | 36.29 | 33.48 | 31.86 | 31.94 | 27.45 | 33.29 |
| | SSIM | 0.0961 | 0.9404 | 0.9163 | 0.8945 | 0.8385 | 0.8361 | 0.6625 | **0.9623** |

Table 20: Comparison of KHINR and various INR baselines using PSNR and SSIM metrics on the CHL Dataset, evaluated across diverse sampling ratios and multiple tasks.

| Task | Metric | SIREN | FFN-P | FFN-G | MMGN | CMLP | MFN | WIRE | KHINR |
|------|--------|-------|-------|-------|------|------|-----|------|-------|
| | | | | 10% sampling | | | | | |
| Task 1 | PSNR | 16.78 | 14.25 | 16.20 | 15.29 | 17.64 | 17.38 | 12.51 | **18.40** |
| | SSIM | 0.6767 | 0.6453 | **0.6941** | 0.6104 | 0.5591 | 0.5531 | 0.2406 | 0.6212 |
| Task 2 | PSNR | 16.72 | 14.29 | 15.86 | 15.62 | 17.47 | 17.17 | 12.50 | **18.33** |
| | SSIM | 0.6680 | 0.6403 | **0.6960** | 0.5868 | 0.5277 | 0.5382 | 0.2794 | 0.6179 |
| Task 3 | PSNR | 20.56 | 21.13 | 20.32 | 20.20 | 18.28 | 19.45 | 12.91 | **21.37** |
| | SSIM | 0.7267 | 0.7686 | 0.6994 | 0.6924 | 0.5552 | 0.6126 | 0.2563 | **0.8140** |
| Task 4 | PSNR | 20.53 | **21.09** | 20.34 | 19.93 | 18.84 | 19.44 | 12.85 | 20.52 |
| | SSIM | 0.7250 | 0.7677 | 0.7013 | 0.6776 | 0.5975 | 0.6120 | 0.2534 | **0.7862** |
| | | | | 25% sampling | | | | | |
| Task 1 | PSNR | 19.75 | 17.53 | 19.34 | 18.98 | 18.82 | 18.97 | 12.90 | **20.27** |
| | SSIM | 0.7353 | 0.7178 | 0.7203 | 0.7173 | 0.6118 | 0.5994 | 0.2796 | **0.7493** |
| Task 2 | PSNR | 19.19 | 17.50 | 18.87 | 18.71 | 18.52 | 18.65 | 12.58 | **19.88** |
| | SSIM | 0.7166 | 0.7175 | 0.7011 | 0.6755 | 0.5972 | 0.5867 | 0.2336 | **0.7382** |
| Task 3 | PSNR | 21.09 | 21.53 | 20.72 | 20.60 | 18.60 | 19.68 | 13.69 | **23.80** |
| | SSIM | 0.7537 | 0.7803 | 0.7222 | 0.7155 | 0.5789 | 0.6287 | 0.2910 | **0.8647** |
| Task 4 | PSNR | 20.85 | 21.44 | 20.69 | 20.30 | 19.30 | 19.62 | 12.86 | **22.30** |
| | SSIM | 0.7407 | 0.7797 | 0.7202 | 0.6979 | 0.6261 | 0.6232 | 0.2526 | **0.8335** |
| | | | | 50% sampling | | | | | |
| Task 1 | PSNR | 20.92 | 20.31 | 20.51 | 20.48 | 19.07 | 19.58 | 13.92 | **22.83** |
| | SSIM | 0.7669 | 0.7780 | 0.7274 | 0.7328 | 0.6145 | 0.6284 | 0.3223 | **0.8596** |
| Task 2 | PSNR | 20.43 | 19.89 | 20.12 | 19.94 | 19.02 | 19.35 | 12.79 | **22.25** |
| | SSIM | 0.7501 | 0.7604 | 0.7227 | 0.6965 | 0.6174 | 0.6188 | 0.2433 | **0.8399** |
| Task 3 | PSNR | 21.36 | 21.74 | 20.97 | 20.76 | 19.36 | 19.76 | 13.41 | **24.77** |
| | SSIM | 0.7670 | 0.7879 | 0.7309 | 0.7250 | 0.6303 | 0.6337 | 0.2864 | **0.8806** |
| Task 4 | PSNR | 21.23 | 21.69 | 20.93 | 20.47 | 19.20 | 19.78 | 12.90 | **23.67** |
| | SSIM | 0.7609 | 0.7881 | 0.7302 | 0.7079 | 0.6172 | 0.6355 | 0.2550 | **0.8613** |

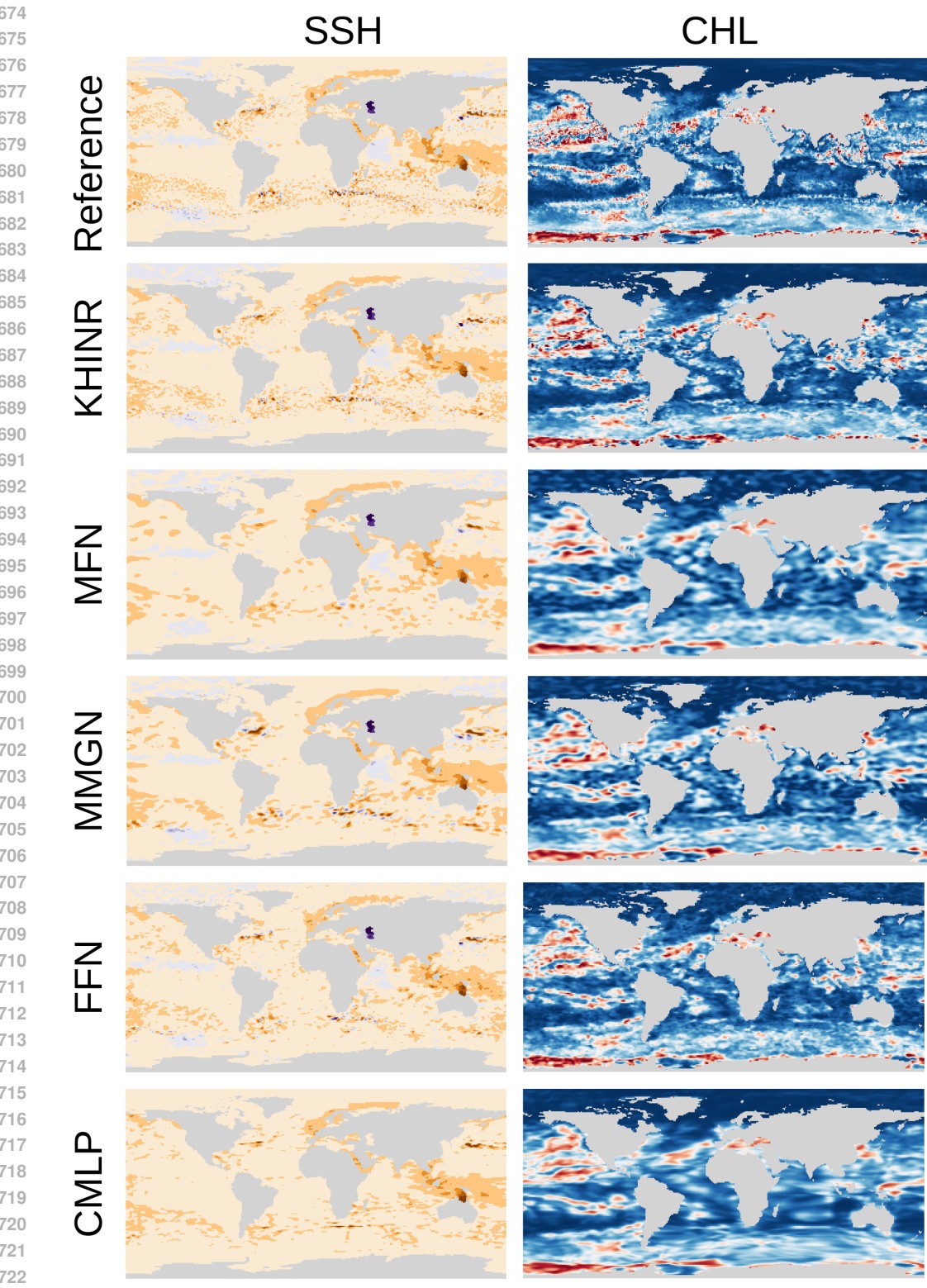

Figure 8: Qualitative Results: Sample reconstructed fields of SSH and CHL by KHINR (Ours) and other leading INRs for Task 1 and 25% sampling ratio. We see that KHINR is able to reconstruct features with higher fidelity than all other models.

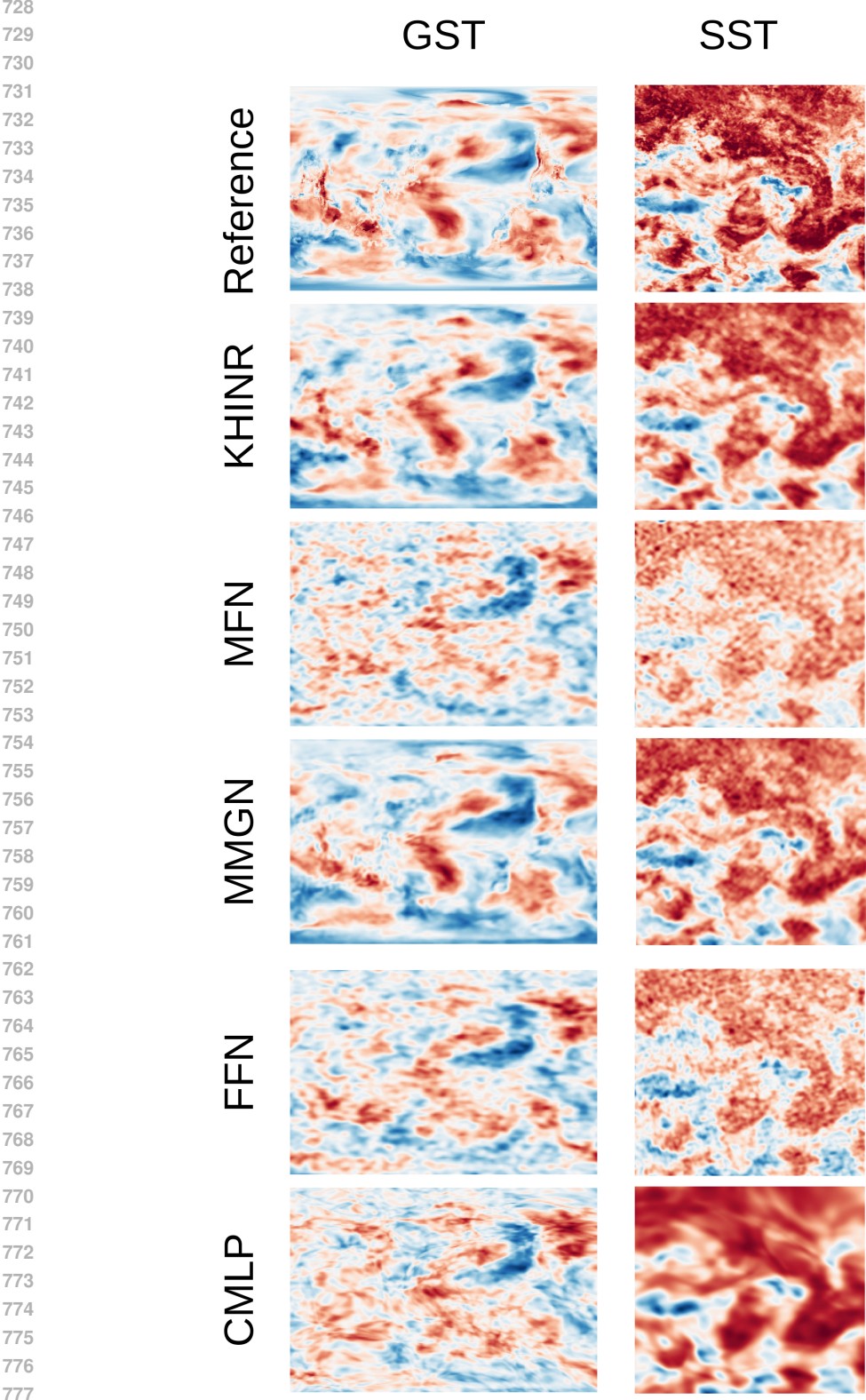

Figure 9: Qualitative Results: Sample reconstructed fields of GST and SST by KHINR (Ours) and other leading INRs for Task 4 (0.3% sampling ratio for SST, 5% sampling ratio for GST) at the time of writing. We see that KHINR is able to reconstruct features with higher fidelity than all other models.

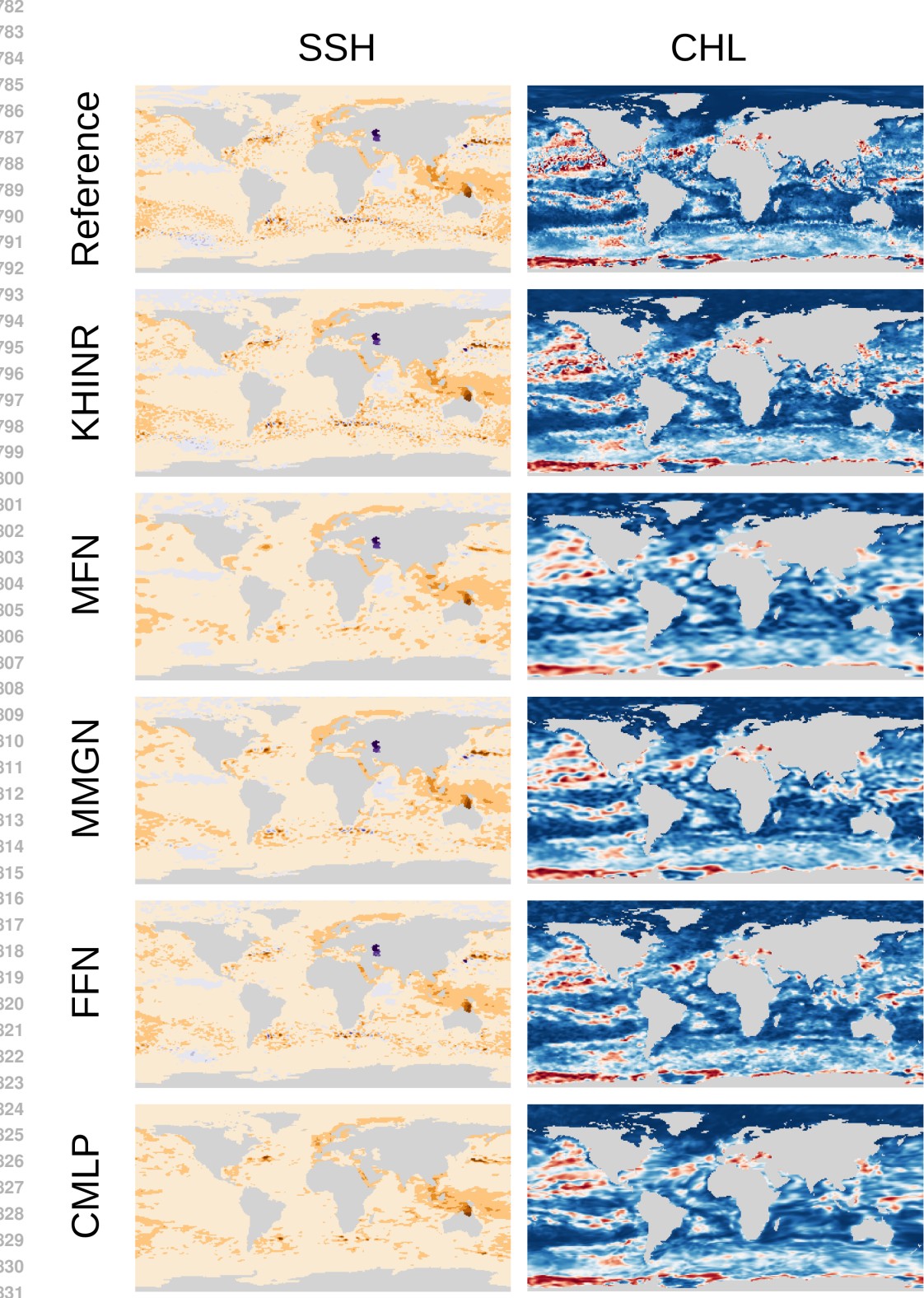

Figure 10: Qualitative Results: Sample reconstructed fields of SSH and CHL by KHINR (Ours) and other leading INRs for Task 3 and 25% sampling ratio. We see that KHINR is able to reconstruct features with higher fidelity than all other models.

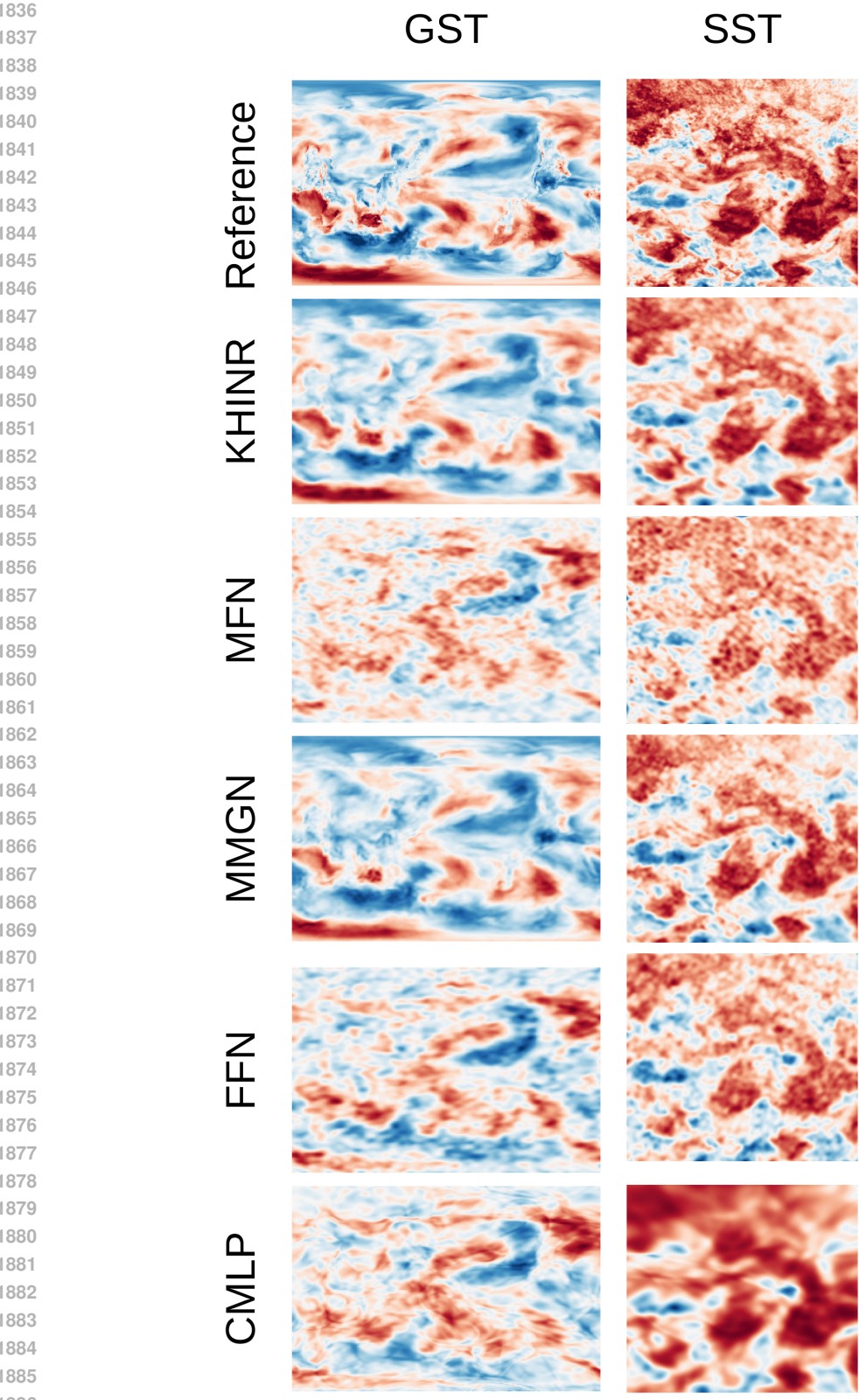

Figure 11: Qualitative Results: Sample reconstructed fields of GST and SST by KHINR (Ours) and other leading INRs for Task 3 (0.3% sampling ratio for SST, 25% sampling ratio for GST) at the time of writing. We see that KHINR is able to reconstruct features with higher fidelity than all other models.

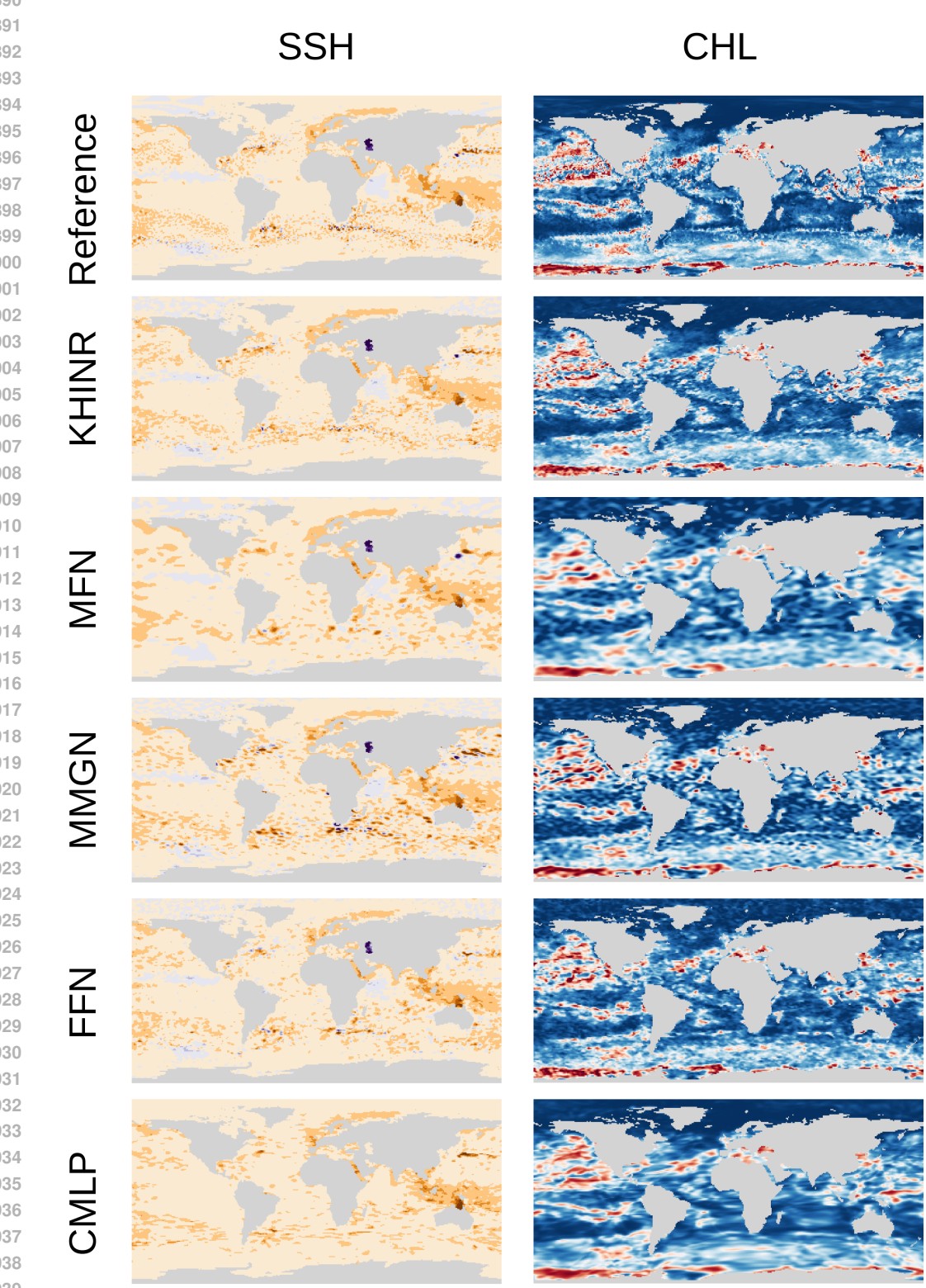

Figure 12: Qualitative Results: Sample reconstructed fields of SSH and CHL by KHINR (Ours) and other leading INRs for Task 2 and 25% sampling ratio. We see that KHINR is able to reconstruct features with higher fidelity than all other models.

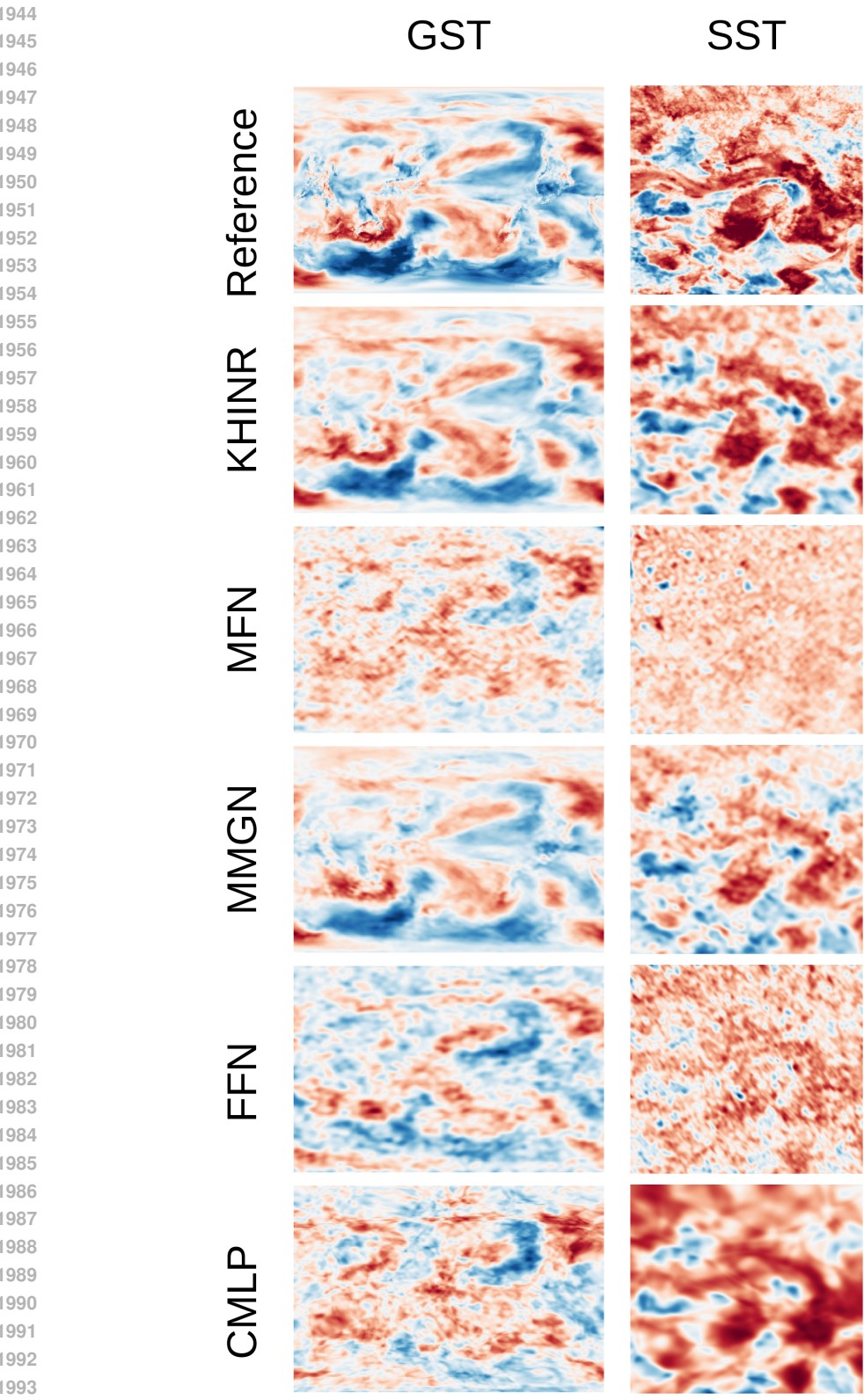

Figure 13: Qualitative Results: Sample reconstructed fields of GST and SST by KHINR (Ours) and other leading INRs for Task 2 (0.3% sampling ratio for SST, 25% sampling ratio for GST) at the time of writing. We see that KHINR is able to reconstruct features with higher fidelity than all other models.

