# OpenReview forum: "Kolmogorov-Arnold Hierarchical Implicit Neural Representation Model for Physical Field Reconstruction"
_ICLR.cc/2026/Conference — Submitted to ICLR 2026_

### Official Review · Reviewer_z15k · 2025-10-21

**Soundness:** 2
**Presentation:** 2
**Contribution:** 2
**Rating:** 4
**Confidence:** 3

**Summary:**

This paper presents KHINR, a hierarchical implicit neural representation (INR) model for reconstructing continuous physical fields from sparse and irregular data. To address the spectral bias, global basis dependence, and limited multiscale modeling of existing INRs, KHINR incorporates learnable Gabor filters for localized frequency-aware encoding, hierarchical Kolmogorov–Arnold Network (KAN) blocks for capturing complex multiscale dependencies, and a latent cross-attention mechanism for efficient global structure learning.  Evaluated on diverse datasets, KHINR outperforms other baselines. Ablation studies validate the contributions of each component.

**Strengths:**

1. The paper has a clear and logical structure.

2. Experiments show good performance and robustness across diverse datasets and settings.

**Weaknesses:**

1. This paper is also closely related to several very recent studies, such as [R1],  **which is a INR-based approach for physics**; [R2][R3][R4] is about diffusion-based methods for physical field reconstruction.
[R1] Wang, Honghui, et al. "GridMix: Exploring Spatial Modulation for Neural Fields in PDE Modeling." The Thirteenth International Conference on Learning Representations. 2025.
[R2] Du, Pan, et al. "Conditional neural field latent diffusion model for generating spatiotemporal turbulence." Nature Communications 15.1 (2024): 10416.
[R3] Chen, Panqi, et al. "Generating Full-field Evolution of Physical Dynamics from Irregular Sparse Observations." The Thirty-ninth Annual Conference on Neural Information Processing Systems.2025.
[R4] Li, Zeyu, et al. "Learning spatiotemporal dynamics with a pretrained generative model." Nature Machine Intelligence 6.12 (2024): 1566-1579.
But the authors do not discover them.


2. The contribution appears  largely incremental by combing KANs within INRs (all existing techniques).  The proposed  latent cross-attention mechanism seems not new.


3. The baselines are not comprehensive, as they mainly include methods from the computer vision field rather than approaches specifically designed for physical field reconstruction.

**Questions:**

1. Figure 2 and its corresponding description (i.e., Section 2.3) are very vague. I cannot match them up. It is unclear how the data flows and how the model operates during training and testing. Providing a clearer mathematical formulation would help, and it would also be useful to highlight the novelty of the proposed method, if any.

2. The proposed architecture appears similar to Senseiver (Santos et al., 2023), which also employs cross-attention to query points for sparse field reconstruction. Since the authors have already mentioned this work, it would be appropriate to include a comparison with it.


3. The reference entitled "Continuous field reconstruction from sparse observations with implicit neural networks" is cited twice. Please check if they are the same.

---

> ### Author Response · Authors · 2025-11-21
>
> ***Weakness***
>
> **Q1: This paper is also closely related $\cdots$ do not discover them.**
>
> A1: We thank the reviewer for bringing GridMix (ICLR 2025) and other references to our attention. We have further carefully considered each of these methods and strengthened our work accordingly.
>
> *Key Problem Setup Distinction:* A critical difference between our work and methods [R2], [R3], and [R4] is the training data requirement. Our problem specifically focuses on *learning from sparse observations only* during training (as stated in Section 2.1), whereas:
>
> 1. CoNFiLD [R2] requires full-field turbulent flow data during training
>
> 2. SDIFT [R3] requires complete spatiotemporal field sequences for learning the diffusion prior
>
> 3. S3GM [R4] first trains an unconditional generative model on full-field data, then conditions on sparse sensors
>
> This fundamental requirement stems from how diffusion models learn: they are trained to reverse a noise-corruption process by learning to denoise complete data samples. The diffusion process gradually adds noise to full-field ground truth data, and the model learns to reverse this process through denoising score matching or variational objectives on complete fields. Without access to the full-field data distribution during training, the generative prior cannot be properly established. While these methods can condition on sparse observations at test time (after the prior is learned), they fundamentally require dense ground truth during the training phase to learn the underlying field distribution. Furthermore, all those references operate on dataset sizes that are much smaller than the full scale reconstruction that we are attempting.
>
> In many real-world scenarios (e.g., environmental monitoring with limited sensor networks), complete field data is simply unavailable during the training phase, making these approaches infeasible. However, we have acknowledged their relevance and included them in our Related Work (Appendix B) given the similarity in reconstruction objectives.
>
> *Added Comparisons:* In response to reviewer feedback, we have significantly expanded our baseline comparisons in the revised manuscript to include
>
> 1. FINER and INCODE: Added to Table 16 (SSH and GST datasets)
>
> 2. FNO, DeepONet, U-Net: Added to Table 15 as non-INR baselines
>
> 3. GridMix [R1]: *We are actively conducting the remaining experiments for this model. Due to computational constraints, the results will be reported at the earliest opportunity in the rebuttal phase.*
>
> *Summary of results:* As shown in Tables 15-16, KHINR consistently achieves the lowest MSE on all tasks and sampling ratios, demonstrating superior performance even when compared to this expanded set of strong baselines.
>
> We believe these additions provide a more comprehensive evaluation and better contextualize our contributions within the broader landscape of field reconstruction methods.

---

> > ### Author Response · Authors · 2025-11-21
> >
> > ***Weakness***
> >
> > **Q2: The contribution $\cdots$ seems not new.**
> >
> > A2: We appreciate the reviewer's concern regarding novelty. While our work does build upon existing components (KANs and INRs), we respectfully request the reviewer to reconsider their assessment as our contribution is much more that a straightforward combination as explained below.
> >
> > *Architectural Novelty (clarified in revised Section 2):* In response to reviewer feedback, we have substantially revised Section 2 to provide clearer exposition of each architectural component and its design rationale.
> >
> > 1. *Hierarchical KAN Block (Section 2.3.2):* Our hierarchical structure enables multi-scale feature learning, achieving 20-36\% error reduction over single-block and sequential configurations (Table 3). This design choice, where lower layers capture fine-grained patterns and higher layers extract semantic relationships, is novel to INR-based reconstruction.
> >
> > 2. *Gabor-Enhanced Latent Cross-Attention (Section 2.3.1)*: Specifically designed for sparse reconstruction, this mechanism reduces training time by up to 50\% while maintaining quality (Figure 3, Table 11) a critical advantage for large-scale scientific applications.
> >
> > 3. *Gating Mechanism (Section 2.3.3)*: Integrates coordinate-based, latent, and temporal information through non-linear fusion.
> >
> > We have also performed comprehensive empirical validation on par with the ICLR community standards. Briefly, our empirical evaluation includes:
> >
> > 1. 13 baselines in INR and non-INR methods (Tables 1-2, 15-16). In comparison, previous papers such as MMGN (ICLR 2024) only compare against 6 baselines.
> >
> > 2. Complete ablations of all components (Section 4, Appendix F.4)
> >
> > 3. Sensitivity analysis on hyperparameters (Tables 12-14)
> >
> > 4. 48 experimental conditions (4 datasets $\times$ 4 tasks $\times$ 3 sparsity levels). These datasets represent a cross section of real situations faced in physical field reconstruction and are twice the number of experiments in MMGN (ICLR 2024).
> >
> > 5. Statistical validation with uncertainty (Table 10)
> >
> > *On Cross-Attention:* While attention mechanisms are common, our Gabor-enhanced latent cross-attention specifically addresses sparse physical field reconstruction by combining localized adaptive frequency representations with computational efficiency. This combination is absent in previous work.
> >
> > The revised Section 2 now explicitly articulates these design rationales and novelties, demonstrating that our contribution extends meaningfully beyond a simple combination of components.
> >
> > **Q3: The baselines are not comprehensive $\cdots$ physical field reconstruction.**
> >
> > A3: Thank you for recommending more baselines. In our revised manuscript, we have significantly expanded our baseline comparisons to include both non-INR methods from scientific ML and additional INR-based approaches:
> >
> > Non-INR Scientific ML Baselines (Table 15): Fourier Neural Operator (FNO), DeepONet, U-Net
> >
> > Additional INR Baselines (Table 16): Senseiver (sparse field reconstruction with cross-attention), FINER (flexible spectral-bias tuning), INCODE (prior knowledge embeddings)
> >
> > Our evaluation now encompasses 13 total baselines: 4 non-INR methods (FNO, DeepONet, U-Net, MMGN) and 9 INR-based methods (SIREN, FFN-P, FFN-G, CMLP, MFN, WIRE, Senseiver, FINER, INCODE). As shown in Tables 1-2, 15-16, KHINR consistently outperforms all baselines under all experimental conditions.
> >
> > ***Questions***
> >
> > **Q1: Figure 2 $\cdots$ method, if any.**
> >
> > A1: We sincerely apologize for the very short methodology section that we included in the original manuscript due to space constraints that led to this question. We also believe that much of the weakness stated above is due to the short methodology section. We thank the reviewer for the opportunity to revise the methodology section. Please consider our revised version while making your recommendation. We are happy to answer further questions.
> >
> > **Q2: The $\cdots$ comparison with it.**
> >
> > A2: Thank you for the suggestion. We have added a comparison with the Senseiver in our revised version (Table 16). Briefly, the mechanism of operation of Senseiver and KHINR (Ours) is different, and we have expanded Section 2 to highlight the exact steps in our approach and refined our Related Work section with more baselines.
> >
> > **Q3: The reference entitled $\cdots$ they are the same.**
> >
> > A3: Apologies for the inconvenience; both refer to the same paper, but we utilized different versions in our bib file. We corrected this in the revised version.

---

> > > ### Comment · Reviewer_z15k · 2025-11-27
> > >
> > > Thanks to the authors for their efforts in preparing the response and conducting the additional experiments. The new results further support the effectiveness of the proposed method.
> > >
> > > However, the current revisions reads more like a laboratory experiment report rather than a scientific research manuscript, and it is far from meeting the  standards expected at ICLR. I strongly encourage the authors to study how high-quality papers present their methods in a coherent and progressive manner with professional notations.

---

> > > > ### Author Response · Authors · 2025-11-27
> > > >
> > > > We thank the reviewer for highlighting this issue and for recognizing the strength of our additional experiments. We understand that the current revision may appear overly procedural. This structure was introduced after several reviewers explicitly requested a clearer, stepwise explanation of our architecture. Our intention was to address those concerns and enhance the presentation without diminishing its scientific rigor. Your previous feedback was valuable in enhancing our revised version. We remain open to constructive feedback that can further strengthen the manuscript. We respectfully request you to reconsider the recommendation.

---

### Official Review · Reviewer_cvbA · 2025-10-27

**Soundness:** 2
**Presentation:** 2
**Contribution:** 2
**Rating:** 4
**Confidence:** 4

**Summary:**

The paper introduces KHINR, a hierarchical implicit neural representation that reconstructs continuous spatio-temporal fields from sparse observations. It integrates hierarchical KAN blocks for multi-scale inductive bias, learnable Gabor encoding for localized frequency representation, and latent cross-attention with gating for efficient global–local fusion. Evaluated on four geophysical datasets under various sparsity regimes, KHINR consistently outperforms INR baselines in accuracy and efficiency, with ablations showing that KAN and Gabor modules drive performance while latent attention improves training efficiency.

**Strengths:**

1. Directly targets INR pain points in scientific data: spectral bias, anisotropy/locality, and global dependency modeling.
2. Hierarchical KAN provides multi-scale, interpretable inductive bias and consistently outperforms MLP/sequential KAN.
3. Localized Gabor encoding adapts to spatially varying frequencies/directions, with notable gains at extreme sparsity (e.g., SST 0.1%).
4. Comprehensive task suite (fixed/random number & locations) and systematic ablations.

**Weaknesses:**

1. The paper lacks sufficient implementation details regarding key architectural components. In particular, the gating mechanism and the way the temporal variable t is incorporated into the model are not clearly explained.
2. Baseline coverage: Missing comparisons to strong non-INR baselines common in scientific ML (e.g., FNO, U-Net on gridded data, Kriging/GP, DeepONet/Neural Operators, low-rank/kernel methods). Current evidence is mainly intra-INR.
3. Reproducibility gaps:
3.1. Exact Gabor parameterization: μ/γ/W/b initialization, constraints, and regularization;
3.2. Latent cross-attention/gating specification (heads, scaling, norm, residuals, complexity);
3.3. KAN spline order, coefficient regularization;

**Questions:**

1.	What is the precise cross-attention/gating formulation? How are Q/K/V constructed between latent features and Gabor-encoded coordinates? Multi-head? Residual/norm?
2.	Gabor parameters: Do learned μ cluster around non-stationary hotspots? Any regularization to prevent collapse? How do you bound γ to balance locality/globality?
3.	The paper states that the Hierarchical KAN captures “multi-scale structures”. Could the authors elaborate on how these scales are defined and implemented? For example, do different KAN sub-blocks operate on distinct frequency ranges, or receptive field sizes?
4.	Temporal handling: If time is present, is it encoded like space or as a conditioning variable?
5.	Could you add comparisons to FNO, U-Net (gridded), Kriging/GP, DeepONet/Koopman or include some in the appendix?

---

> ### Author Response · Authors · 2025-11-21
>
> ***Weakness***
>
> **Q1: The paper lacks $\cdots$ are not clearly explained.**
>
> A1: We apologize for the brief methodology section in the earlier submission (due to space constraints) that led to this weakness. We appreciate and agree with your observation to improve our methodology section. We have now revised Section 2.3 to incorporate a detailed explanation of the gating mechanism and temporal variable encoding.
>
> **Q2: Baseline coverage: $\cdots$ mainly intra-INR**
>
> A2: We thank the reviewer for this valuable feedback regarding baseline coverage. We acknowledge that our original submission did not sufficiently emphasize a critical aspect of our problem setting.
>
> *Clarification on problem formulation:* According to our problem statement, full field data is not accessible during the training phase. We train solely on sparse observations, making INRs particularly well-suited for this setting. Although the problem statement reflected this constraint, we apologize that it was not made sufficiently explicit. We have now strengthened both the problem statement (Section 2.1) and the abstract to clearly emphasize this key training requirement that distinguishes our work from methods requiring dense ground truth during training.
>
> *Added non-INR baselines:* To strengthen our evaluation, we now compare KHINR against three prominent non-INR architectures commonly used in scientific ML (FNO, DeepONet, UNet). We adapted grid-based approach to our sparse setting by taking sparse locations as input and computing loss only at available sparse locations during training matching our training protocol exactly. Results are provided in Table 15 (Appendix G.1.1).
>
> **Q3: Reproducibility gaps: $\cdots$ coefficient regularization**
>
> A3: We thank the reviewer for identifying these gaps. Once again we apologize for the brief methodology section. We have substantially revised Section 2 to address all concerns.  Gabor parameterization is given in the new Section 2.2.2; Latent cross-attention/gating are in new Sections 2.3.1, 2.3.3. Complete hyperparameter configurations in Table 6 (Appendix E.1), training details in Section 3.2. Code is also provided for reproducibility.
>
> ***Questions***
>
> **Q1: What is the precise $\cdots$ Residual/norm?**
>
> A1: We have significantly revised Section 2 to address this question and others, We now provide complete mathematical details of architecture components including:
>
> 1. Gabor Enhanced Latent Cross-attention specification (Section 2.3.1)
>
> 2.  Hierarchical KAN blocks (Section 2.3.2)
>
> 3. Gating mechanism (Section 2.3.3):
>      We have added a complete four-stage description with mathematical formulations,
>
>       Stage 1: Latent construction with channel-wise averaging and Gabor modulation
>
>       Stage 2: Block-wise fusion of coordinate, latent, and temporal terms with explicit equations for each
>
>       Stage 3: Multiplicative fusion across blocks with design rationale
>
>       Stage 4: Final MLP mapping
>
> We request the reviewer to reconsider their recommendation on the basis of the revised manuscript. We strongly believe that our contribution addresses a timely problem with strong architectural innovations, exhaustive empirical validation and most importantly the best physical field reconstruction quality (Figure 1 and qualitative results).
>
> **Q2: Gabor parameters: $\cdots$ locality/globality?**
>
> A2: We thank the reviewer for the question about Gabor parameter behavior and regularization.
>
> *Clustering of $\boldsymbol{\mu}$ around hotspots:* Yes, the learned Gabor centers $\boldsymbol{\mu}$ naturally adapt to regions of high spatial variability through gradient-based optimization. We believe that during training, filters whose centers align with informative regions (e.g., sharp gradients, boundaries, or non-stationary patterns) receive stronger gradients and thus learn more discriminative features. Though we do not enforce any constraints, it explicitly emerges from the reconstruction objective.
>
> *Regularization to prevent collapse:* We do not use explicit regularization terms (e.g., diversity losses) on $\boldsymbol{\mu}$ or $\gamma$. However, collapse is prevented through several implicit mechanisms: (1) We initialize $\boldsymbol{\mu} \sim \mathcal{N}(2,1)$ across the spatial domain; (2) And $\gamma \sim Gamma(\alpha, \beta)$ (as noted in Section 2.2.2); (3) The reconstruction loss naturally encourages diversity. If entire concentration of filter collapse to same regions, they become redundant and fail to capture the full signal complexity, resulting in higher loss.
>
> *Bounding $\gamma$ for locality/globality balance:* We regulate $\gamma$ using $\alpha$ and $\beta$ as $\gamma \sim Gamma(\alpha, \beta)$ which helps to controls the spatial extent of Gabor filter.

---

> > ### Author Response · Authors · 2025-11-21
> >
> > ***Questions***
> >
> > **Q3: The paper $\cdots$ receptive field sizes?**
> >
> > A3: We thank the reviewer for this important question about multi-scale structure capture in hierarchical KAN.
> >
> > Scale emergence: Multi-scale capability emerges naturally during training rather than being explicitly assigned. Our K hierarchical KAN blocks process the latent representation L, each learning different functional transformations through their univariate functions. During training, different blocks naturally specialize: some learn smooth, slowly-varying transformations capturing large-scale patterns (e.g., global circulation), while others learn rapidly-varying transformations capturing fine-scale structures (e.g., sharp gradients, eddies). This specialization occurs because the multiplicative gating mechanism rewards complementary rather than redundant features.
> >
> > Hierarchical processing: Within each block, stacked layers create a processing hierarchy where lower layers handle fine-grained features and higher layers extract abstract representations. This is analogous to CNNs but through learned univariate function compositions rather than fixed architectural constraints.
> >
> > Effective receptive fields: Different blocks develop different effective receptive fields through their learned transformations. Blocks with smooth functions integrate information over larger spatial regions, while blocks with sharper functions operate locally. The Gabor-enhanced latent representation provides spatial context, and each block's functions determine how much context influences its output.
> >
> > Empirical validation: Table 3 shows hierarchical KAN outperforms single-block and sequential configurations by 12-36\%. This superior performance confirms different blocks capture complementary scales if all blocks learned identical representations, no performance gain would occur.
> >
> > **Q4: Temporal handling: $\cdots$ conditioning variable?**
> >
> > A4: We are representing time as a conditional variable distinct from locations. A new section 2.3.3 and 2.3.4 in the revised version gives the description.
> >
> > **Q5: Could you add comparisons to FNO, U-Net (gridded), Kriging/GP, DeepONet/Koopman or include some in the appendix?**
> >
> > A5: Thank you for your suggestion. We have now included non-INR based FNO, U-Net and DeepONet results in Appendix G.1.1 in the revised version.
> >
> > We also included additional INR Baselines (Table 16): Senseiver (sparse field reconstruction with cross-attention), FINER (flexible spectral-bias tuning), INCODE (prior knowledge embeddings)
> >
> > Our evaluation now encompasses 13 total baselines: 4 non-INR methods (FNO, DeepONet, U-Net, MMGN) and 9 INR-based methods (SIREN, FFN-P, FFN-G, CMLP, MFN, WIRE, Senseiver, FINER, INCODE). As shown in Tables 1-2, 15-16, KHINR consistently outperforms all baselines under all experimental conditions.
> >
> > Note on Kriging/GP and kernel methods: These classical approaches face severe computational challenges at the scale of our datasets (e.g., SST at 0.01 degree spatial resolution). The $O(N^3)$ complexity of standard GP inference makes them infeasible for our large-scale reconstruction tasks.

---

### Official Review · Reviewer_Xja4 · 2025-10-31

**Soundness:** 2
**Presentation:** 2
**Contribution:** 2
**Rating:** 6
**Confidence:** 4

**Summary:**

This paper proposes KHINR (Kolmogorov–Arnold Hierarchical Implicit Neural Representation), a novel model for reconstructing continuous physical fields from sparse sensor data. KHINR consists of three components. [1] Hierarchical Kolmogorov–Arnold Networks (KANs) – to model nonlinear, multiscale relationships via learnable univariate spline functions. [2] Learnable Gabor filters – replacing Fourier features to provide localized, adaptive, frequency-aware spatial encodings. [3]Latent cross-attention mechanism – to fuse sparse observations with global structural context efficiently. Geophysical datasets were tested, and another INR-based approach was compared.

**Strengths:**

[1] The experimental results demonstrate strong performance, even under sparse data conditions.

[2] Multiscale capability: The hierarchical structure effectively captures fine-to-coarse dependencies, which is crucial for modeling complex physical phenomena.

[3] The proposed method is novel, and the introduction of Gabor-enhanced latent attention further improves efficiency by reducing training time.

**Weaknesses:**

[1] Other types of Baselines: The paper focuses primarily on physics-based field reconstruction using an INR-based approach. However, comparisons with other state-of-the-art methods are missing. In particular, it would be valuable to compare this approach with Fourier Neural Operator (FNO), Physics-Informed Neural Networks (PINNs), and diffusion-based models,.

[2] Variety of Experiments: All experiments are conducted on 2D static fields, without addressing spatiotemporal coupling or dynamic PDE constraints. It would be important to discuss whether the proposed method can generalize to 3D datasets or to time-dependent problems in 2D problem, as this would better demonstrate its scalability and versatility.

[3] Model Efficiency: KAN-based architectures typically require longer training times and involve more parameters compared to MLP. While the Appendix reports testing time, there is no detailed analysis of training efficiency, parameter count, or computational complexity (e.g., GFLOPs). The training time is reported in Figure 3, but a comprehensive comparison of training times with other methods would provide a better understanding.  Such an analysis would provide a more thorough understanding of the model’s efficiency.

[4] Ablation and Variants: The Hierarchical KAN block appears to be a core contribution of the paper, yet no ablation studies are provided to analyze the effect of different configurations (e.g., different numbers of KAN layers). Moreover, with the recent introduction of KAN 2.0 [r1], it would be interesting to investigate whether replacing the existing KAN layers with KAN 2.0 improves performance or training stability.[r1] Liu, Ziming, et al. "Kan 2.0: Kolmogorov-arnold networks meet science." arXiv preprint arXiv:2408.10205 (2024).

**Questions:**

See the Weakness for the questions.

---

> ### Author Response · Authors · 2025-11-21
>
> ***Weakness***
>
> **Q1: Other type $\cdots$ and diffusion-based models**
>
> A1: Thank you for your suggestion. We have incorporated non-INR based results in Appendix G.1.1 of the revised version, and a discussion on comparing with diffusion models is included in the response to Reviewer z15k.
>
> **Q2: Variety $\cdots$ its scalability and versatility**
>
> A2: Thank you for this valuable comment. We fully agree that extending our framework to spatiotemporal settings and 3D domains is an important direction. Our current study focuses on 2D static fields to isolate and evaluate the core contribution learning robust reconstructions under extreme sparsity without conflating additional sources of complexity such as high-dimensional PDE structure. We strongly believe that 2D reconstruction is of great value to the community as evidenced by several other papers that we have included as references. However, please do note that the proposed architecture is dimension-agnostic: the Gabor parameterization, sparse sensor encoding, and reconstruction pipeline can be readily extend to 3D inputs  with minimal architectural changes (primarily in the coordinate representation). We are already exploring these extensions, and plan to include results on 3D volumetric fields as part of our future work.
>
> **Q3: Model Efficiency $\cdots$ the model’s efficiency**
>
> A3: For a thorough computational analysis, please refer to Appendix F.1. The KHINR model achieves the top rank while maintaining similar inference times and model size, thus demonstrating superior efficiency and performance compared to other baseline models. Additionally, in Figure 3, we have illustrated the training time with and without Gabor-enhanced latent cross attention, which underscores one of our contribution to field reconstruction.
>
> **Q4: Ablation and Variants: $\cdots$ arXiv preprint arXiv:2408.10205 (2024).**
>
> A4: In Appendix F.4.3, we previously presented results with different numbers of KAN layers. Additionally, we provided outcomes with varying hidden and latent dimensions, as well as the impact of our novel components discussed in Section 4, Model Analysis, in the main paper.
>
> In our problem formulation, the input follows the structure (B, N, D), where B is the batch size, N corresponds to the sequence length (sparse or dense spatial points), and D denotes the feature dimension. We also explored the suggested KAN 2.0 implementation during our experimentation phase; however, since it is designed for 2D inputs, it did not align with the requirements of our setting.

---

> > ### Comment · Reviewer_Xja4 · 2025-11-27
> >
> > Thanks for the reviewer's response, which addresses the majority of my questions and concerns.  Therefore, I will maintain my original rating.

---

### Official Review · Reviewer_zSNB · 2025-10-31

**Soundness:** 3
**Presentation:** 1
**Contribution:** 3
**Rating:** 4
**Confidence:** 3

**Summary:**

The paper proposes a method to address the sparse-to-dense physical field
reconstruction problem, i.e., how to recover the complete physical field when only a few
sensor measurements are available. The authors argue that existing INRs perform poorly
on this specific task because conventional MLP-based INRs suffer from spectral bias and
a poor inductive bias when modeling multiscale, anisotropic geophysical fields.
Moreover, global positional encodings do not adapt well to spatially varying frequencies
and orientations. Additionally, there is no efficient mechanism to inject sparse
observations into a global latent representation. The proposed KHINR model addresses
these issues by incorporating hierarchical KAN blocks to model multiscale structures
more effectively than standard MLPs. It further employs latent cross-attention between
sparse value–location tokens and global latent tokens, followed by a gating mechanism to
align latent codes back to dense coordinates.

**Strengths:**

1). The proposed architecture is carefully designed, rather than being a simple modification of an existing method. The intelligent combination of three powerful components hierarchical KANs, learnable Gabor filters, and latent cross-attention is
highly novel within the scientific machine learning domain.
2). The proposed method (KHINR) obtains superior results across all datasets and all sampling tasks compared to existing methods

**Weaknesses:**

1). The paper is poorly written, and many parts are difficult to understand. I suggest that the authors rewrite the methodology section, as several important concepts are unclearly or inadequately explained. I feel the authors may need to pay more attention to how they present their methodology, as the proposed method and its architecture are only briefly discussed. If the authors could include their design rationale explaining how and why each component was chosen the methodology section would be significantly strengthened.


2). I would like to know what the exact gating mechanism is, as it is barely discussed in the paper.


3). The compared INR baselines are old. I suggest authors to compare their method with recent methods like FINER/INCODE.


4). Many of the recent INRs have not been cited.


5). I would like to know how you conducted the experiments on, for instance, WIRE. Did the authors fine-tune the activation parameters? If so, over what range were they tuned, and were they adjusted for each specific sparsity level?


6). I would like to know How does KHINR differ from INR-based image inpainting? I feel both are highly correlated.


7). The methodology section defines the target function u(x) over a spatiotemporal domain), and the Figure 2 explicitly(disconnectedly) indicates a time input (t). So, I would like to know, how do you handle this time variable? Do the authors use some encoding? No information regarding this is presented.


8). This is a novel research direction with carefully designed architecture. So, I’m willing to accept the paper, if the authors specifically refine their writing part (at least the methodology) and include the latest comparisons.

**Questions:**

Please see the weaknesses section

---

> ### Author Response · Authors · 2025-11-21
>
> ***Weakness***
>
> **Q1: The paper $\cdots$ significantly strengthened.**
>
> A1: We apologize for the brief methodology section. We have now rewritten the methodology section. We thank the reviewer wholeheartedly for recognizing that contribution is worthwhile and allowing us to update the writing.
>
> **Q2: I would $\cdots$ in the paper.**
>
> A2: Sorry for the lack of clarity. In the revised methodology section, we now have an expanded explanation of the gating mechanism. We provide details of each component we have introduced, highlighting their significance.
>
> **Q3,Q4: The comapred $\cdots$  methods like FINER/INCODE. Many of the recent $\cdots$ cited.**
>
> A3,A4: Thank you for your suggestion. We have now added results from INCODE (WACV 2024), FINER (CVPR 2024), SENSEIVER (NMI 2023), and additional Non-INR based baselines in our revised version. While several methods show reasonable performance, none exceed the well-established ICLR 2024 baseline (MMGN). Please refer to G.1.1 and G.1.2 in the Appendix.
>
> Briefly, the results are as follows. Table 15 and 16 in the Appendix G report the new results. KHINR consistently achieves the lowest MSE across all tasks, outperforming INCODE, FINER, and SENSEIVER by a significant margin. The poor performance of SENSEIVER is attributed to its grid-like reconstruction patterns, as illustrated in Figure 2 of the SENSEIVER paper. In fact, none of these models also cross the second best model MMGN (or FFN) that we have included in the main paper. Furthermore, no model is able to match the qualitative reconstruction of our method (Figure 1 and additional results in the Appendix).
>
> **Q5: I would like to know $\cdots$ specific sparsity level?**
>
> A5: Yes, we did a good faith best effort approach for baselines. For baseline methods, we performed a systematic hyperparameter search. Table 6 in the manuscript lists the complete set of activation parameters and training hyperparameters that we tuned for each method. We selected the best-performing configuration from these predefined ranges.
>
> **Q6: I would like $\cdots$ are highly correlated.**
>
> A6: Thank you for raising this question. While KHINR and INR based image inpainting share the high level idea of representing signals with implicit neural representations, their objectives and operating regimes are fundamentally different.
>
> KHINR is designed for physical field reconstruction, where the goal is to recover a continuous field (e.g., fluid velocity, temperature, pressure) from sparse and irregularly sampled physical measurements. The method must respect the structure of scientific data, handle non-uniform sampling, and generalize across a range of sampling ratios and physical reconstruction tasks. Moreover, during training, only the sparse data are available, and most importantly, the full field is not available for the model to learn from. In our experiments, we demonstrated KHINR across multiple physical fields and varying sparse sampling regimes to highlight this generality.
>
> In contrast, INR based image inpainting focuses on filling missing pixel regions of natural images. These methods typically assume structured image masks, dense pixel grids, and rely on local texture/semantic consistency rather than physical coherence or PDE driven behavior.
>
> Thus, although both approaches use INRs, KHINR targets a broader and challenging reconstruction setting that is specific to scientific and physical domains, not traditional image inpainting. This distinction is also reflected in our experimental setup and evaluation design.
>
> **Q7: The methodology section $\cdots$ this is presented.**
>
> A7: We apologize for the confusion. We have now revised Section 2 for improved clarity where we specifically call out each part of Figure 2. We have now provided a clear explanation for time encoding in the Gating mechanism section and Temporal Handling (Section 2.3.4) in the Methodology.
>
> **Q8: This is a novel research direction $\cdots$ include the latest comparisons.**
>
> A8: Thank you for your constructive feedback and for recognizing the efforts in this work. In response, we substantially revised the methodology section, providing clearer mathematical formulations and detailing the role of each architectural component. We have also expanded our experimental comparisons by adding the suggested INR baselines as well as strong non-INR baselines (from other reviewers) in the Appendix to ensure a more comprehensive evaluation and highlight the superior performance of our KHINR. Please let us know if there are any additional analyzes or clarifications that would further strengthen the paper, we would be happy to incorporate them.

---

### Author Response · Authors · 2025-11-26
**General comment on the revised version**

Dear Reviewers,

Thank you for taking the time to review our work. We believe the clarifications and revisions we have provided address the concerns you raised. We would be grateful if you could reconsider your assessment in light of these changes.

Once again we highlight that we achieve excellent physical field reconstruction from sparse observations, beating the recent state-of-the-art models both qualitatively and quantitatively in challenging situations across 48 experimental conditions (4 datasets, 4 tasks, 3 sparsity levels) with diverse baselines (INR and Non-INR). The mechanisms of model operation (including an expanded methodology section) and ablations show evidence of the core contributions.

Should any concerns remain, we welcome further discussion and are happy to provide additional revisions as needed.

---

### Author Response · Authors · 2025-12-01
**Summary of Contributions and Reviewer Discussion**

Dear Area Chairs,

**Contributions:**

1. We have developed a novel architecture in which we introduce the representational strength of Hierarchical Kolmogorov-Arnold Networks (KANs) within INRs for physical field reconstruction, achieving state of the art reconstruction. Evidence is presented in qualitative (Figure 1) and quantitative (Table 1,2) results and in ablation studies (Table 3 and Table 4).

2. Gabor Enhanced latent cross-attention mechanism is introduced to compress global structural dependencies into a lower-dimensional embedding space, allowing scalable reconstruction and maintaining crucial physical relationships across scales. Training time is also reduced (Ablation Table 5; Figure 3).

3. We achieve excellent physical field reconstruction from sparse observations, beating the recent state-of-the-art models both qualitatively and quantitatively in challenging situations across 48 experimental conditions (4 datasets, 4 tasks, 3 sparsity levels) (Table 1 and Table 2) with diverse baselines (Additional results based on the review suggestion: Non-INR (Table 15) and INR (Table 16)). Mechanisms of model operation (including an expanded methodology section) and ablation studies show evidence of the core contributions. Our extensive empirical validation uses $2\times$ experiments and $\sim$ $3\times$ baselines compared to the most recent physical field reconstruction paper published in ICLR 2024 (MMGN).

**Summary of Reviewers Discussion:**

Most of the concerns of the reviewers’ focused on the methodology section and the inclusion of additional baselines. During the rebuttal phase, we thoroughly responded to all of these concerns by expanding the methodology section with more detailed and clearer explanations of each term (which we had previously omitted due to space limitations) and by including new experiments using INR (Table 15) and NON-INR (Table 16) baselines in the revised manuscript. After rewriting the methodology and incorporating these additional baseline results, the reviewers had indicated that they were willing to accept the paper.

---

### Meta-Review · Area_Chair_hkAW · 2026-01-06

**Summary:**

This paper proposes KHINR for sparse-observation physical field reconstruction using hierarchical KAN blocks, learnable Gabor encoding, and latent cross-attention with gating. Reviewers generally agree the problem is important and the empirical results are promising. However, multiple reviewers consistently raised major concerns about clarity and reproducibility: key mechanisms (gating, temporal handling, attention formulation) and experimental protocol details are under-specified, making it difficult to verify what is actually trained and evaluated. While the rebuttal adds stronger baselines and claims a rewritten methodology, the manuscript still remains hard to understand and the evaluation setup remains insufficiently transparent for confident acceptance.

**Reviewer Concerns:**

Addressed by rebuttal (partially):

Baseline coverage was expanded. The authors added several newer INR baselines (e.g., FINER/INCODE/Senseiver) and some non-INR baselines (e.g., FNO/U-Net/DeepONet), which improves coverage compared to the initial submission.
The authors stated they rewrote the methodology and added more details on gating and temporal handling.
Still outstanding (major):

1.	Clarity and method specification remain inadequate. Multiple reviewers report that Figure/Section describing the core pipeline is hard to match, and key components (gating, temporal handling, attention formulation, tensor shapes/notation) are still not presented in a clear, consistent, and verifiable way. One reviewer explicitly noted the revised manuscript still reads like a procedural lab report rather than an ICLR-quality research paper with professional notation and coherent presentation.

2.	Experimental protocol is under-specified, harming reproducibility. The paper does not clearly specify train/validation/test splitting (especially in time), Task2/Task4 sampling distributions and resampling frequency, and whether tuning is performed per sparsity level. This makes it hard to assess potential leakage and to reproduce results.

3.	Fairness of newly added baselines is not fully established. While the authors claim to adapt non-INR methods to sparse supervision by computing loss only on observed points, the adaptation details (input construction, masking/interpolation choices, tuning budget alignment) are not sufficiently described.

4.	Novelty is difficult to judge without clear specification. Given the remaining ambiguity in the method and protocol, it is hard to precisely determine the contribution beyond combining known components, as pointed out by reviewers.

**Reviewer Scores:**

Reviewer zSNB (4): Likely unchanged. Rebuttal adds comparisons and claims improved explanation, but clarity and protocol details remain a major concern.

Reviewer Xja4 (6): Likely unchanged (maintains weak accept). The rebuttal addresses many of their questions, and they already indicated they would keep the original rating.

Reviewer cvbA (4): Likely unchanged. Added details and baselines help, but key implementation/notation details and reproducibility gaps are still not fully resolved.

Reviewer z15k (4): Likely unchanged. They acknowledged the additional experiments support effectiveness, but strongly criticized the revised manuscript’s presentation and notation quality as below ICLR standards.

---

### Decision · Program_Chairs · 2026-01-26

Reject